# 🥑 AVoCaDO: An Audiovisual Video Captioner Driven by Temporal Orchestration

**Xinlong Chen**[2,3,1]*, **Yue Ding**[2,3], **Weihong Lin**[1], **Jingyun Hua**[1], **Linli Yao**[4], **Yang Shi**[4],
**Bozhou Li**[4], **Qiang Liu**[2,3]†, **Yuanxing Zhang**[1], **Pengfei Wan**[1], **Liang Wang**[2,3]

[1]Kling Team, Kuaishou Technology
[2]New Laboratory of Pattern Recognition (NLPR),
Institute of Automation, Chinese Academy of Sciences (CASIA)
[3]School of Artificial Intelligence, University of Chinese Academy of Sciences
[4]Peking University

**Project webpage:** https://avocado-captioner.github.io/

## Abstract

Audiovisual video captioning aims to generate semantically rich descriptions with temporal alignment between visual and auditory events, thereby benefiting both video understanding and generation. In this paper, we present **AVoCaDO**, a powerful AudioVisual video Captioner Driven by the temporal Orchestration between audio and visual modalities. We propose a two-stage post-training pipeline: (1) **AVoCaDO SFT**, which fine-tunes the model on a newly curated dataset of 107K high-quality, temporally-aligned audiovisual captions; and (2) **AVoCaDO GRPO**, which leverages tailored reward functions to further enhance temporal coherence and dialogue accuracy while regularizing caption length and reducing collapse. Experimental results demonstrate that AVoCaDO significantly outperforms existing open-source models across four audiovisual video captioning benchmarks, and also achieves competitive performance on the VDC and DREAM-1K benchmarks under visual-only settings.

## 1 Introduction

In the era of multimodal large language models (MLLMs), video captioning plays a critical role in advancing video understanding. In addition to facilitating the alignment of multimodal representations during pretraining (Xu et al., 2021; Li et al., 2024), it also functions as a key mechanism for injecting semantic knowledge into downstream video understanding and generation tasks (Sun et al., 2019; Hong et al., 2022; Zhang et al., 2025b). Recent studies (Chen et al., 2024; 2025c; Zhang et al.; Wang et al., 2025b) have shown that training with higher-quality video captions not only improves captioning performance, but also yields consistent improvements across a broad spectrum of downstream applications. Therefore, advancing the capabilities of video captioning models offers a foundational pathway toward building more powerful video understanding and generation systems.

Despite notable progress in recent video captioning models (Xu et al., 2024; Chai et al., 2024; Yuan et al., 2025; Shi et al., 2025b; Ren et al., 2024; Shen et al., 2023), most existing approaches remain predominantly vision-centric, often overlooking the rich semantic cues embedded in audio signals. In practice, auditory elements, such as dialogues, voiceovers, and background music, are indispensable for achieving a holistic and contextually grounded understanding of video content. A truly comprehensive captioning model should therefore integrate and reason jointly over both visual and auditory modalities. A common workaround for vision-only models is to generate an independent audio caption via a separate audio model and concatenate it to the visual description. However, such a decoupled strategy inherently fails to model fine-grained temporal alignment and causal interplay between audiovisual events, limiting its reliability in practical applications.

---

*This work was conducted during the author's internship at Kling Team, Kuaishou Technology
†Corresponding author: qiang.liu@nlpr.ia.ac.cn

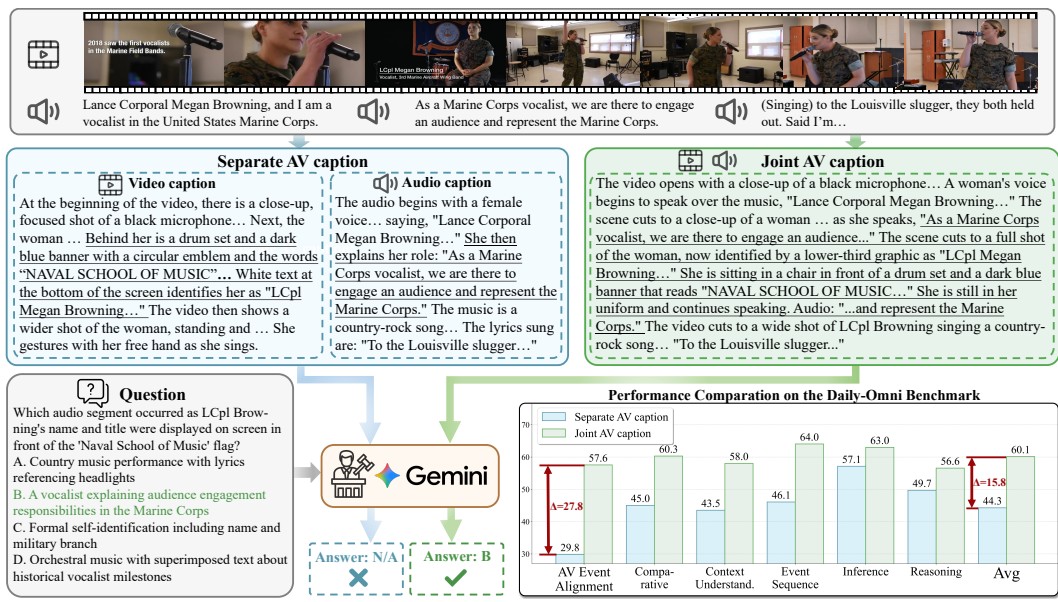

Figure 1: Schematic illustration of the pilot experiment. In this example, naively concatenating captions from the video and audio modalities fails to yield a correct answer to the corresponding question. In contrast, jointly processing both modalities to generate a time-aligned caption provides sufficient information, as indicated by the underlined text.

To validate the importance of audiovisual alignment, we conduct a pilot experiment on Daily-Omni (Zhou et al., 2025). Using Gemini-2.5-Pro (Comanici et al., 2025), we generate two types of captions: one by processing visual and audio inputs separately and then concatenating their resulting captions; and the other by jointly processing both modalities to produce a temporal-aligned caption. A judge model (also Gemini-2.5-Pro) is then tasked with answering questions based solely on the textual captions. As shown in Fig. 1, the joint approach yields a significant performance improvement, with an average accuracy gain of 15.8%. This gap is even more pronounced in the "AV Event Alignment" category, where it reaches 27.8%, underscoring the critical necessity of audiovisual temporal alignment in captions for comprehensively understanding the video content.

Based on the above analysis, we propose **AVoCaDO**, an audiovisual video captioner that effectively integrates visual and auditory events in temporal synchrony. Built upon Qwen2.5-Omni (Xu et al., 2025a), which aligns visual and audio signals via interleaved token sequences, AVoCaDO is further enhanced through a two-stage post-training pipeline: (1) AVoCaDO SFT, where we collect and construct a dataset of 107K high-quality audiovisual video-caption pairs for supervised fine-tuning, with particular emphasis on temporal alignment between visual and audio events during caption generation; (2) AVoCaDO GRPO, where we introduce a reward function based on key event alignment to optimize the temporal coherence of audio and visual information. Additionally, we design two auxiliary rewards to further enhance dialogue accuracy, reduce repetition collapse and regulate caption length. Collectively, these optimizations tailor AVoCaDO to generate captions that are not only semantically rich but also temporally aligned with audiovisual inputs. Extensive experiments demonstrate that AVoCaDO significantly outperforms existing open-source models across multiple audiovisual captioning benchmarks, and achieves competitive performance on the VDC Detailed subset (Chai et al., 2024) and DREAM-1K (Wang et al., 2024), which evaluate captions in visual-only settings. Our contributions can be summarized as follows:

- We propose AVoCaDO, a powerful audiovisual video captioner that effectively integrates visual and auditory events with a strong emphasis on temporal alignment. This model will be open-source to facilitate future research in more video understanding and generation tasks.

- We design a two-stage post-training pipeline for AVoCaDO: (1) AVoCaDO SFT, leverages a 107K high-quality audiovisual caption dataset to enhance temporal alignment; and (2) AVoCaDO

GRPO, which employs carefully designed reward functions to improve temporal coherence and dialogue accuracy while regularizing caption length and reducing collapse.

- Extensive experiments show that AVoCaDO outperforms all existing open-source audiovisual models and even surpasses the commercial Gemini-2.5-Pro on UGC-VideoCap (Wu et al., 2025). It also achieves competitive performance under visual-only settings.

## 2 RELATED WORKS

### 2.1 VIDEOLLMS FOR VIDEO CAPTIONING

Recent advances in Video Large Language Models (VideoLLMs) (Zhang et al.; OpenBMB, 2025; Zhang et al., 2025a; Shi et al., 2025a;c) have substantially enhanced progress in video captioning. These VideoLLM-based captioners (Ren et al., 2025; Xue et al., 2025; Yao et al., 2024) typically employ a video encoder to capture video semantics and then bridge them with an LLM to generate high-quality captions. To further describe fine-grained video cues, OwlCap (Zhong et al., 2025) and Tarsier series (Wang et al., 2024; Yuan et al., 2025) construct large-scale, high-quality SFT datasets to enable the generation of detailed captions that balance dynamic motion and static detail.

However, most of these efforts are vision-centric, while neglecting audio content, which plays a vital role in forming a comprehensive understanding of video content. Although several recent audiovisual VideoLLMs (Cheng et al., 2024; Geng et al., 2025; Liu et al., 2025b; Sun et al., 2024; Hua et al., 2025) have incorporated both modalities, they are not specifically optimized for the captioning task. Concurrent to our work, video-SALMONN-2 (Tang et al., 2025) and UGC-VideoCaptioner (Wu et al., 2025) have also explored audiovisual video captioning. Nevertheless, the former requires computationally intensive post-training involving six rounds of DPO with sample pairs selected solely based on atomic event metrics, while the latter is limited to short-form user-generated videos. In contrast, our AVoCaDO achieves precise temporal alignment of audiovisual events through a relatively lightweight training process guided by more holistic audiovisual considerations, and is capable of generating temporally synchronized, high-quality captions across diverse scenarios.

### 2.2 REINFORCEMENT LEARNING FOR VIDEOLLMS

Reinforcement Learning (RL) (Christiano et al., 2017) has attracted increasing attention in VideoLLMs for enhancing complex reasoning through explicit thinking chains and verifiable reward designs. Video-R1 (Feng et al., 2025b), VerIPO (Li et al., 2025c), and LongVILA-R1 (Chen et al., 2025d) adopt GRPO (Shao et al., 2024) with rule-based rewards to improve performance on general video understanding tasks. Similarly, Time-R1 (Wang et al., 2025c), TAR-TVG (Guo et al., 2025), and Tempo-R0 (Yue et al., 2025) introduce IoU-related rewards to advance temporal grounding.

However, these task-specific approaches are not well-suited for detailed video captioning. Verifying long video descriptions remains challenging, as they are prone to visual *omissions* and *hallucinations*. At present, only a few RL-based methods explicitly target video captioning. VideoChat-R1 (Li et al., 2025b) leverages event-recall rewards to improve caption quality. VersaVid-R1 (Chen et al., 2025a) balances the accuracy and completeness of captions through a meticulously designed reward mechanism. VideoCap-R1 (Meng et al., 2025) decomposes captioning into structured thinking and caption generation stages, integrating thinking and captioning scorers to improve output quality.

In summary, these studies focus on only specific aspects of visual-only captioning. By contrast, our work proposes a holistic reward design to enhance temporal coherence and dialogue accuracy while regularizing caption length and reducing collapse, which is tailored for audiovisual video captioning, resulting in significant gains in fine-grained caption quality across multiple dimensions.

## 3 AVoCaDO

AVoCaDO is powered by a carefully designed post-training pipeline tailored specifically for audiovisual video captioning. This pipeline consists of two sequential stages: the AVoCaDO SFT stage, followed by the AVoCaDO GRPO stage. We select Qwen2.5-Omni-7B as the base model for its built-in ability to align video frames and audio signals using interleaved token sequencing.

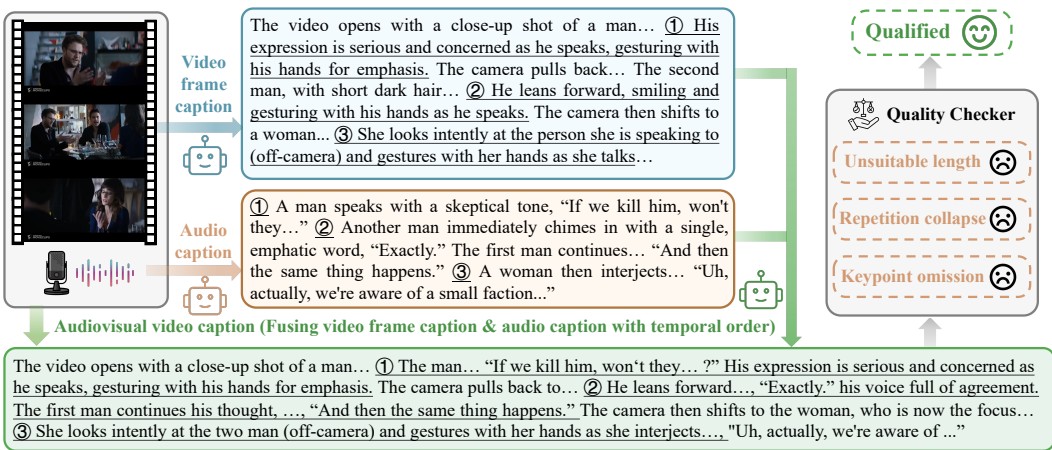

Figure 2: The pipeline for generating high-quality temporally-aligned audiovisual video captions. For clarity, corresponding audio-visual events before and after fusion are marked with circled numbers and underlined for reference.

## 3.1 AVoCADO SFT

In this stage, we train the base model using 107K high-quality audiovisual video-caption pairs curated by us. The dataset is constructed by collecting videos from diverse sources and pairing them with meticulously generated captions. The curation procedure is described below.

To enhance the model's capability in describing complex audiovisual interactions, we collect short-form videos rich in auditory elements, including mixed speech, background music, and sound effects. Specifically, we source 24K videos from TikTok-10M (Company, 2025) and 18K from Short-Video (Shang et al., 2025), both of which offer dense, real-world audiovisual scenarios ideal for audiovisual understanding. To further strengthen the model's grasp of multi-scene spatio-temporal dynamics and cinematic transitions, we randomly sample 20K videos from Shot2Story (Han et al., 2023). Additionally, we incorporate 29K samples from FineVideo (Farré et al., 2024), 11K from YouTube-Commons (Pierre-Carl, 2024), and 5K from CinePile (Rawal et al., 2024) to further improve the model's generalization performance across diverse audiovisual contexts.

Although the pilot experiment confirms the importance of audiovisual joint captioning, we observe that directly generating such joint captions may sometimes lead to information omissions from either the audio or visual stream (see App. D.1 for details). To obtain semantically rich and temporally aligned captions, we adopt a two-stage captioning strategy, as illustrated in Fig. 2. First, we utilize Gemini-2.5-Pro to generate modality-specific captions separately from the video frames and the audio track. These separate captions, along with the original video, are then fed back into Gemini-2.5-Pro to be synthesized into a temporally coherent multimodal caption by aligning events across modalities according to the temporal structure of the video. Finally, a quality checker is employed to ensure high data quality. Initially, clearly low-quality captions, such as those with inappropriate length or repetitive patterns, are filtered out. The remaining samples then undergo a completeness assessment, where both the pre- and post-synthesis captions are presented to GPT-4.1[1] for scoring on a 1–5 scale based on synthesis completeness. Only samples scoring 4 or above are retained, thereby reducing the risk of critical information loss during multimodal fusion.

## 3.2 AVoCADO GRPO

To further enhance the model's capabilities in audiovisual video captioning, we adopt the Group Relative Policy Optimization (GRPO) algorithm (Shao et al., 2024), training the model on a randomly selected subset of 2K samples from our SFT dataset. As shown in Fig. 3, we design three complementary reward functions to guide the optimization process: (1) a checklist-based reward that promotes comprehensive coverage of audiovisual keypoints; (2) a dialogue-based reward that

---

[1] https://platform.openai.com/docs/models/gpt-4.1

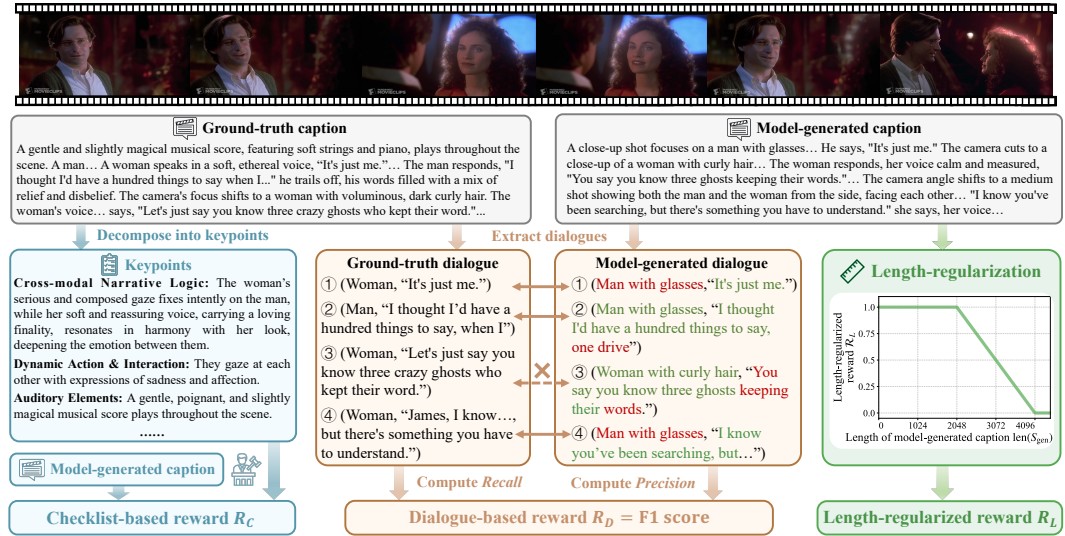

Figure 3: Illustration of the three rewards $\mathcal{R}_C$, $\mathcal{R}_D$, and $\mathcal{R}_L$, which are specifically designed for improving the quality of audiovisual video captioning.

targets the ASR fidelity and speaker identification accuracy of dialogues, a critical component of audiovisual content; and (3) a length-regularized reward that mitigates repetition collapse and regulates caption length. These reward functions complement each other and work synergistically to optimize various critical aspects for enhancing the overall captioning quality.

### 3.2.1 GROUP RELATIVE POLICY OPTIMIZATION

GRPO significantly reduces both training time and GPU memory usage by eliminating the need for a separate critic model in Proximal Policy Optimization (PPO). Specifically, GRPO works by sampling a group of $G$ responses $\{o_1, o_2, ..., o_G\}$ for each question $q$ from the old policy model $\pi_{\theta_{old}}$, then computing their corresponding rewards $\{r_1, r_2, ..., r_G\}$ to derive the advantage function $A_i$ for response $o_i$:

$$A_i = \frac{r_i - \text{mean}(\{r_1, r_2, \ldots, r_G\})}{\text{std}(\{r_1, r_2, \ldots, r_G\})} \tag{1}$$

The current policy model $\pi_\theta$ is then optimized using the following objective function:

$$\begin{aligned}
\mathcal{J}_{\text{GRPO}}(\theta) = \mathbb{E}_{\{o_i\}_{i=1}^G \sim \pi_{\theta_{\text{old}}}(o_i|q)} \Bigg[ \frac{1}{G} \sum_{i=1}^G \bigg( &\min\Big(\frac{\pi_\theta(o_i|q)}{\pi_{\theta_{\text{old}}}(o_i|q)} A_i, \\
&\text{clip}\Big(\frac{\pi_\theta(o_i|q)}{\pi_{\theta_{\text{old}}}(o_i|q)}, 1-\varepsilon, 1+\varepsilon\Big) A_i\Big) - \beta \cdot \mathbb{D}_{\text{KL}}\left(\pi_\theta || \pi_{\text{ref}}\right) \bigg) \Bigg],
\end{aligned} \tag{2}$$

### 3.2.2 CHECKLIST-BASED REWARD

To enhance the overall completeness of audiovisual video captioning, we propose a checklist-based reward $\mathcal{R}_c$ grounded in fine-grained content decomposition. Specifically, each ground-truth caption $S_{\text{gt}}$ is pre-decomposed by GPT-4o into a structured inventory of keypoints $K = \{k_1, k_2, \ldots, k_n\}$, with $n = |K|$ indicating the inventory size. These keypoints are organized according to a comprehensive taxonomy spanning five core dimensions tailored to audiovisual caption:

- **Cross-modal Narrative Logic:** High-level coherence in which auditory and visual modalities mutually explain, complement, or guide each other to reveal underlying intent or storyline; explicit temporal alignment between modalities is required.
- **Dynamic Action & Interaction:** Motions, events, and pairwise or group interactions among entities, capturing the evolving narrative dynamics of the scene.

- **Auditory Elements:** All sound-related content, including speech, music, and ambient or diegetic sound effects, which is essential for holistic multimodal comprehension.
- **Spatio-temporal & Cinematography:** Structural and stylistic features, such as scene transitions, temporal progression, and camera techniques that shape perceptual and narrative flow.
- **Static Entity Description:** Attributes and spatial configurations of relatively stationary entities, including persons, objects, and environmental elements.

During GRPO training, for a generated caption $S_{\text{gen}}$, the checklist-based reward $\mathcal{R}_c$ is defined as:

$$\mathcal{R}_c(S_{\text{gen}} \mid K) = \frac{1}{|K|} \sum_{i=1}^{|K|} \text{Judge}(S_{\text{gen}}, k_i) \tag{3}$$

where $\text{Judge}(S_{\text{gen}}, k_i) \in \{0, 1\}$ is the matching score assigned by a judge model, specifically, GPT-4.1, indicating whether $S_{\text{gen}}$ correctly mentions keypoint $k_i$.

### 3.2.3 DIALOGUE-BASED REWARD

In parallel, dialogue serves as a critical component of audiovisual content. Therefore, we further design a dialogue-based reward $\mathcal{R}_D$ to ensure the ASR fidelity and speaker identification accuracy of a dialogue in captions.

As shown in Fig. 3, we first extract and structure dialogues from captions as a list using Gemini-2.5-Pro, where each entry consists of a speaker and their corresponding spoken content. Let the model-generated dialogue sequence be denoted as $D_{gen} = \left[(s_1^{gen}, c_1^{gen}), (s_2^{gen}, c_2^{gen}), \dots, (s_N^{gen}, c_N^{gen})\right]$, and the ground-truth dialogue sequence as $D_{gt} = \left[(s_1^{gt}, c_1^{gt}), (s_2^{gt}, c_2^{gt}), \dots, (s_M^{gt}, c_M^{gt})\right]$, where $s_i^*$ represents the speaker, $c_i^*$ is the spoken content of the $i$-th dialogue unit, and $M$ and $N$ are the lengths of the two sequences, respectively.

To compute $R_D$, we need to simultaneously consider the speaker similarity $S_{\text{speaker}}$ and content similarity $S_{\text{content}}$ between $D_{\text{gen}}$ and $D_{\text{gt}}$. To this end, we adopt a two-step strategy: first, we match dialogue units based on content similarity; then, we verify speaker consistency for the matched pairs.

For any dialogue content pair $\left(c_i^{\text{gen}}, c_j^{\text{gt}}\right)$, where $i \in [1, N]$ and $j \in [1, M]$, their content similarity $\text{Sim}\left(c_i^{\text{gen}}, c_j^{\text{gt}}\right)$ is measured using the edit distance[2] between the two strings, calculated as:

$$\text{Sim}\left(c_i^{\text{gen}}, c_j^{\text{gt}}\right) = 1 - \frac{\text{edit\_distance}\left(c_i^{gen}, c_j^{gt}\right)}{\max\left(\text{len}\left(c_i^{gen}\right), \text{len}\left(c_j^{gt}\right)\right)} \tag{4}$$

where $\text{len}(\cdot)$ denotes the string length. Our goal is to identify a subsequence of dialogue units from $D_{\text{gen}}$ that matches positionally with a subsequence of the same length from $D_{\text{gt}}$, such that the content similarity $\text{Sim}(\cdot)$ of each aligned pair exceeds a predefined threshold $\gamma$, and the total content similarity $S_{\text{content}}$ is maximized.

The search for this optimal subsequence is analogous to the classical Longest Common Subsequence (LCS)[3] problem and can be solved via dynamic programming. Let $F_{i,j}$ represent the maximum total content similarity achievable from the first $i$ dialogue units of $D_{gen}$ and the first $j$ dialogue units of $D_{gt}$. The transition equation is defined as follows:

$$F_{i,j} = \begin{cases} 0 & \text{if } i = 0 \text{ or } j = 0 \\ \max\left\{F_{i-1,j}, F_{i,j-1}\right\} & \text{if } i > 0, j > 0, \text{Sim}\left(c_i^{\text{gen}}, c_j^{\text{gt}}\right) < \gamma \\ \max\left\{F_{i-1,j}, F_{i,j-1}, F_{i-1,j-1} + \text{Sim}\left(c_i^{\text{gen}}, c_j^{\text{gt}}\right)\right\} & \text{if } i > 0, j > 0, \text{Sim}\left(c_i^{\text{gen}}, c_j^{\text{gt}}\right) \geq \gamma \end{cases}$$

where the similarity threshold $\gamma$ is set to 0.6.

After identifying the optimal matched subsequence based on the dialogue content, we further assess speaker consistency (assigned as 0 or 1) for each matched pair based on the video content

---

[2]https://en.wikipedia.org/wiki/Edit_distance
[3]https://en.wikipedia.org/wiki/Longest_common_subsequence

using Gemini-2.5-Pro, and the total number of correctly matched speaker pairs serves as the speaker similarity $S_{\text{speaker}}$. The final similarity $S_{\text{combined}}$ between the two sequences is then calculated as:

$$S_{\text{combined}} = (S_{\text{speaker}} + S_{\text{content}}) \,/\, 2 \tag{5}$$

From a physical interpretation, $S_{\text{combined}}$ represents the proportion of correct dialogue units in $D_{gen}$, which takes values in the range $\big[0, \min(M, N)\big]$. The recall and precision are then computed as:

$$\text{Rec} = S_{\text{combined}} \,/\, M, \quad \text{Prec} = S_{\text{combined}} \,/\, N \tag{6}$$

The final dialogue-based reward $\mathcal{R}_D$ is defined as the F1 score:

$$\mathcal{R}_D = 2 \cdot \text{Prec} \cdot \text{Rec} \,/\, (\text{Prec} + \text{Rec}) \tag{7}$$

### 3.2.4 Length-Regularized Reward

For video captioning, output repetition collapse remains a frequently observed issue (Li et al., 2023; Yao et al., 2025a). Moreover, in practical deployment scenarios, it is essential to balance inference efficiency with caption quality, which often necessitates maintaining moderate output length.

To mitigate the rate of repetition collapse and enhance inference efficiency, we design length-regularized reward $\mathcal{R}_L$ that encourage complete captions while penalizing excessive length. The thresholds $\tau_1$ and $\tau_2$ are set to 2048 and 4096 respectively, which is analyzed in App. D.2.

$$\mathcal{R}_L = \begin{cases} 1.0, & \text{if } \text{len}(S_{\text{gen}}) < \tau_1 \\ 1 - \dfrac{\text{len}(S_{\text{gen}}) - \tau_1}{\tau_2 - \tau_1}, & \text{if } \tau_1 \leq \text{len}(S_{\text{gen}}) < \tau_2 \\ 0.0, & \text{otherwise} \end{cases} \tag{8}$$

During GRPO training, we use the sum of the aforementioned three rewards as the final reward $\mathcal{R}$.

$$\mathcal{R} = \mathcal{R}_C + \mathcal{R}_D + \mathcal{R}_L \tag{9}$$

## 4 Experiments

### 4.1 Experimental Settings

#### 4.1.1 Baselines

First, we consider two concurrent works focusing on audiovisual video captioning, video-SALMONN-2 and UGC-VideoCaptioner, as important baselines. Additionally, we evaluate several popular general-purpose audio-visual understanding models, covering both open-source (Qwen-Omni series (Xu et al., 2025a;b), HumanOmniV2 (Yang et al., 2025), ARC-Hunyuan-Video (Ge et al., 2025), MiniCPM-o-2.6 (OpenBMB, 2025)) and commercial options (Gemini-2.5 series). To further assess the importance of audio modality, we compare against some strong vision-only models, including Qwen2.5-VL (Bai et al., 2025), InternVL3.5 (Wang et al., 2025a).

#### 4.1.2 Benchmarks

For audiovisual video captioning, we evaluate models on video-SALMONN-2 testset, UGC-VideoCap, Daily-Omni and WorldSense (Hong et al., 2025). The former two benchmarks evaluate caption quality directly, while the latter two are originally designed for audiovisual video question-answering (QA). To adapt these QA-oriented benchmarks for caption evaluation, we first use the target model to generate a caption for each video, and then utilize a judge model (Gemini-2.5-Pro) to answer questions solely based on the textual captions. To mitigate answer guessing when the caption lacks necessary information, we instruct the judge model to refrain from answering such questions, which are then marked as incorrect samples. Additionally, we evaluate models on the "detailed" subset of the VDC and DREAM-1K benchmarks under visual-only settings.

---

[4]https://platform.openai.com/docs/models/gpt-3.5-turbo

| Model | Size | Modality | video-SALMONN-2 testset | | | UGC-VideoCap | | | |
|---|---|---|---|---|---|---|---|---|---|
| | | | Miss ↓ | Hall. ↓ | Total ↓ | Audio ↑ | Visual ↑ | Detail ↑ | Avg. ↑ |
| Gemini-2.5-Pro | - | A + V | 18.1 | 13.3 | 31.3 | 69.5 | 74.7 | 73.7 | 72.6 |
| Gemini-2.5-Flash | - | A + V | 19.3 | 13.9 | 33.3 | 69.1 | 75.8 | 74.0 | 73.0 |
| InternVL3.5 | 8B | V | 53.8 | 25.5 | 79.4 | 47.9 | 64.8 | 59.5 | 57.4 |
| Qwen2.5-VL | 7B | V | 40.5 | 17.0 | 57.5 | 46.6 | 69.1 | 62.3 | 59.3 |
| HumanOmniV2 | 7B | A + V | 49.2 | **12.3** | 61.6 | 45.6 | 66.3 | 59.5 | 57.1 |
| ARC-Hunyuan-Video | 7B | A + V | 45.7 | 12.5 | 58.2 | 52.7 | 56.0 | 55.8 | 54.8 |
| Qwen2.5-Omni | 7B | A + V | 41.7 | 15.4 | 57.1 | 46.9 | 66.1 | 60.0 | 57.7 |
| MiniCPM-o-2.6 | 8B | A + V | 42.2 | 14.3 | 56.5 | 38.6 | 68.5 | 57.7 | 54.9 |
| UGC-VideoCaptioner* | 3B | A + V | 31.6 | 17.0 | 48.6 | 61.4 | 58.4 | 57.5 | 59.1 |
| video-SALMONN-2* | 7B | A + V | 21.2 | 17.6 | 38.8 | 61.8 | 71.4 | 68.5 | 67.2 |
| Qwen3-Omni-Instruct | 30B-A3B | A + V | 32.0 | 13.6 | 45.6 | 67.5 | 74.8 | 72.3 | 71.5 |
| Qwen3-Omni-Captioner | 30B-A3B | A + V | 31.0 | 16.6 | 47.6 | 69.0 | 75.5 | 72.3 | 72.5 |
| AVoCaDO (Ours) | 7B | A + V | **21.1** | 16.2 | **37.3** | **73.0** | 74.6 | 71.8 | 73.2 |

Table 1: Model performance on the audiovisual video captioning benchmarks. "A" and "V" refer to the audio and visual modalities, respectively. The results presented above are reproduced using the official code. Note that the video-SALMONN-2 testset originally employed GPT-3.5[4] as the judge model, which occasionally led to misjudgments. To ensure more reliable evaluation, we uniformly replaced it with GPT-4.1. *Concurrent works with us.

## 4.2 EXPERIMENTAL RESULTS

### 4.2.1 DIRECT CAPTION EVALUATION

We first evaluate the audiovisual video captioning performance on the video-SALMONN-2 testset and the UGC-VideoCap benchmark, which employ different metrics to directly assess caption quality. As shown in Tab. 1, our AVoCaDO achieves state-of-the-art performance among all open-source models on both benchmarks.

Notably, while some open-source models, such as HumanOmniV2, exhibit a slightly lower Hallucination rate compared to AVoCaDO on the video-SALMONN-2 testset, this is because these models are not specifically optimized for detailed captioning and tend to produce overly brief descriptions that fail to convey the full content of the video, leading to a significantly higher Miss rate and weaker performance on UGC-VideoCap. In contrast, AVoCaDO strikes a better balance between comprehensiveness and accuracy, ultimately outperforming all open-source models in both the Total metric on the video-SALMONN-2 testset and the average score on UGC-VideoCap.

Moreover, compared to the latest large-scale MoE-based omni model, Qwen3-Omni, AVoCaDO still demonstrates better performance. Remarkably, AVoCaDO even surpasses the Gemini-2.5 series on UGC-VideoCap, highlighting its strong capability in audiovisual video captioning.

| Model | Size | Daily-Omni | World-Sense |
|---|---|---|---|
| Gemini-2.5-Pro | - | 60.2 | 33.8 |
| Gemini-2.5-Flash | - | 55.3 | 31.0 |
| HumanOmniV2 | 7B | 8.2 | 6.6 |
| ARC-Hunyuan-Video | 7B | 8.6 | 8.7 |
| MiniCPM-o-2.6 | 8B | 9.8 | 7.2 |
| Qwen2.5-Omni | 7B | 13.4 | 8.6 |
| UGC-VideoCaptioner | 3B | 17.0 | 11.2 |
| video-SALMONN-2 | 7B | 29.9 | 18.2 |
| Qwen3-Omni-Instruct | 30B-A3B | 17.5 | 12.7 |
| Qwen3-Omni-Captioner | 30B-A3B | 27.2 | 14.1 |
| AVoCaDO (Ours) | 7B | **50.1** | **25.7** |

Table 2: QA performance by Gemini-2.5-Pro based on textual captions. To mitigate answer guessing when the caption lacks necessary information, the model is instructed to refrain from answering such questions, which are then marked as incorrect samples.

### 4.2.2 QA-BASED CAPTION EVALUATION

The Daily-Omni and WorldSense benchmarks feature challenging questions that require comprehension of either one or both modalities, along with their temporal relationships. To assess caption quality, we employ a judge model (Gemini-2.5-Pro) that answers these questions based solely on the

textual captions. To reduce speculative answers when the caption lacks essential information, we instruct the judge model to refrain from answering such questions, which are then marked as incorrect.

As shown in Tab. 2, AVoCaDO significantly outperforms existing open-source models of comparable size, as well as the latest large-scale MoE-based Qwen3-Omni series, achieving performance improvements of 20.2% on Daily-Omni and 7.5% on Worldsense over the strongest baseline models.

| Model | Size | VDC Detailed | | DREAM-1K |
|---|---|---|---|---|
| | | Acc | VDCscore | F1 score |
| VideoLLaMA 3 | 7B | 33.4 | 1.9 | 30.5 |
| ShareGPT4Video | 8B | 35.6 | 1.8 | 19.5 |
| AuroraCap | 7B | 41.3 | 2.2 | 20.8 |
| Qwen2.5-VL | 7B | 44.5 | 2.4 | 30.1 |
| Qwen2.5-Omni | 7B | 39.7 | 2.2 | 31.6 |
| video-SALMONN-2 | 7B | 46.1 | **2.5** | 34.4 |
| AVoCaDO (Ours) | 7B | **47.4** | **2.5** | **35.9** |

Table 3: Model performance on the VDC Detailed subset and DREAM-1K, which evaluate captions in visual-only settings.

Additionally, we further evaluate models on the VDC Detailed subset and DREAM-1K, two benchmarks that are specifically designed to measure the captioning performance for visual-only videos. As reported in Tab. 3, AVoCaDO also demonstrates competitive performance in this setting.

| Model | Reward | | | video-SALMONN-2 testset | | | Daily-Omni by caption | | |
|---|---|---|---|---|---|---|---|---|---|
| | $\mathcal{R}_D$ | $\mathcal{R}_C$ | $\mathcal{R}_L$ | Total ↓ | Dlg. F1 ↑ | RepCol (%) ↓ | Avg. ↑ | Dlg. F1 ↑ | RepCol (%) ↓ |
| Qwen2.5-Omni | – | – | – | 57.1 | 7.1 | 7.1 | 13.4 | 16.9 | 8.1 |
| AVoCaDO-SFT | – | – | – | 41.4 | 74.4 | 3.5 | 48.1 | 73.6 | 5.1 |
| AVoCaDO-SFT-2K* | – | – | – | 43.0 | 74.1 | 2.9 | 48.5 | 74.8 | 5.3 |
| AVoCaDO-GRPO | ✓ | – | – | 41.3 | 76.5 | 2.4 | 49.5 | 76.1 | 6.0 |
| | ✓ | ✓ | – | **37.3** | 75.9 | 3.9 | 49.5 | 75.2 | 4.9 |
| | ✓ | ✓ | ✓ | **37.3** | **76.9** | **0.4** | **50.1** | 76.2 | **1.0** |

Table 4: Ablation study on our post-training pipeline. "Dlg. F1" represents the metric of dialogue quality, computed as in Eq. 7. "RepCol" indicates the ratio of generations exhibiting repetition collapse. AVoCaDO-SFT-2K* refers to the model further fine-tuned on AVoCaDO-SFT using the same 2K samples employed during the GRPO phase.

### 4.2.3 ABLATION STUDIES

In Tab. 4, we conduct an in-depth analysis of each component within our post-training pipeline.

First, the AVoCaDO-SFT stage significantly enhances the model's overall performance across three key dimensions: benchmark scores, dialogue quality, and the reduction of repetition collapse in captions. These improvements are consistent on both the video-SALMONN-2 testset, where captions are evaluated directly, and the Daily-Omni benchmark, which assesses caption quality through a QA task. This uniform improvement underscores the effectiveness of our SFT data construction strategy.

In the AVoCaDO-GRPO stage, incorporating the dialogue-based reward $\mathcal{R}_D$ improves the dialogue F1-score by over 2% on both benchmarks. Additionally, the accuracy on Daily-Omni is also enhanced by 1.4%, which is attributed to the model's improved ability to generate detailed and precise dialogue content for answering specific questions. Concurrently, the checklist-based reward $\mathcal{R}_C$ significantly reduces the total error rate on the video-SALMONN-2 testset, underscoring its effectiveness in capturing key audiovisual events. Finally, the length-regularized reward $\mathcal{R}_L$ not only markedly alleviates repetition collapse but also boosts performance across other metrics, highlighting its dual benefit of ensuring conciseness and quality.

To further validate the contribution of these tailored rewards, we additionally fine-tune AVoCaDO-SFT on the same 2K data used in GRPO, yielding AVoCaDO-SFT-2K. However, the model shows no significant performance gains and even exhibits a notable degradation on the video-SALMONN-2 testset. These results suggest that the performance gains stem from the curated reward functions rather than the data volume, confirming their efficacy in advancing audiovisual captioning.

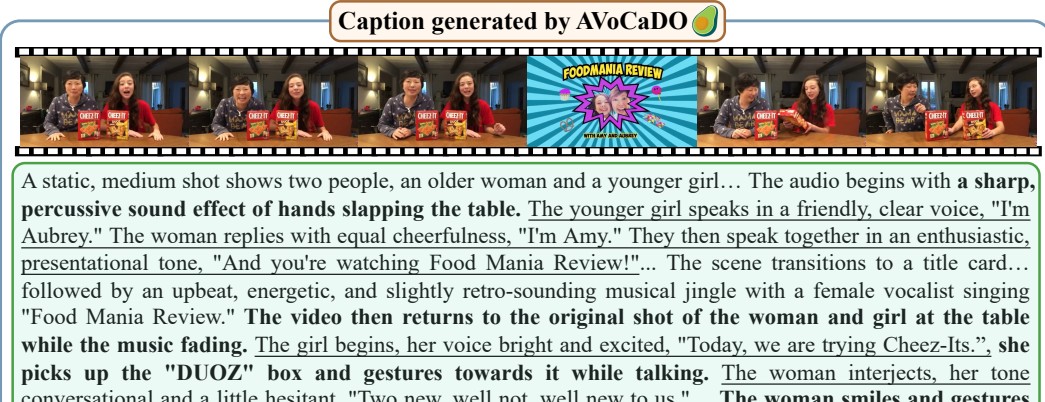

Figure 4: An illustration of a video caption generated by AVoCaDO, featuring both **precise audio-visual temporal alignment** and accurate dialogue rendering.

#### 4.2.4 QUALITATIVE ANALYSIS

Fig. 4 shows a caption generated by AVoCaDO, highlighting its strong capabilities in audiovisual temporal alignment and precise representation of dialogues. More cases can be found in App. F.

## 5 CONCLUSION

This work concentrates on the task of audiovisual video captioning. Initially, we highlight the significant role of temporal alignment between visual and audio events. Informed by this observation, we introduce AVoCaDO, an audiovisual video captioner driven by the temporal alignment between audio and visual modalities. Building upon Qwen2.5-Omni, AVoCaDO is enhanced through a two-stage post-training strategy: AVoCaDO SFT, which fine-tunes the model on a 107K high-quality audiovisual caption dataset emphasizing temporal alignment, and AVoCaDO GRPO, which leverages tailored reward functions to further boost temporal coherence and dialogue accuracy while reducing repetition collapse and regulating caption length. Experimental results demonstrate that AVoCaDO substantially outperforms existing open-source models on four audiovisual video captioning benchmarks and delivers competitive results on the VDC Detailed subset and DREAM-1K, which focus on visual-only video captioning. Ablation studies validate the effectiveness of each component in our training pipeline, underscoring the overall effectiveness of our approach.

## ACKNOWLEDGEMENTS

This work is supported by the Strategic Priority Research Program of Chinese Academy of Sciences (XDA0480102) and National Natural Science Foundation of China (62576339, 92570204, 62236010).

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

# A    DETAILS OF THE TRAINING DATA

The videos used for our training dataset construction come from multiple sources to ensure diverse audiovisual content. Below we provide detailed statistics for each dataset:

- **TikTok-10M** is a large-scale dataset containing 10 million short-form posts from TikTok. The dataset reflects authentic patterns of modern short-form videos, including diverse visual styles, short durations, rich background music and voiceovers, and a wide variety of themes such as entertainment, dance, humor, beauty, and pets. From the full dataset, we select 24K videos for our model training, ensuring a representative coverage of content, audio-visual styles, and user-generated characteristics.

- **Shot2Story** is a dataset comprises 43K multi-shot videos. The length of each video is ranging from 10s to 40s. 20K videos are chosen from the dataset. Each video in the dataset contains multiple shots. This rich multi-shot structure allows our audiovisual caption model to learn to capture key events in each shot and associate them together.

- **ShortVideo** is also a large-scale video dataset from short-video platform including 153,561 videos. These videos have varying durations, ranging from under 30 seconds to over 5 minutes, with most being less than one minute. We randomly choose 18k videos from the dataset for training our model.

- **FineVideo** is a dataset with 43K videos that span 3.4K hours. The videos in the dataset are carefully filtered to retain dynamic content with both visual actions and mid-fast pace spoken language by word density filtering and visual dynamism filtering methods. We select 29K videos from this dataset.

- **YouTube-Commons** is a collection of audio transcripts of 2,063,066 videos shared on YouTube under a CC-By license. The corpus is multilingual, with English as the majority language, and provides automatic translations into several languages such as French, Spanish, German, Russian, Italian, and Dutch. Each video is accompanied by detailed provenance information, including title, link, channel name, and upload date, ensuring transparency and reusability. We sample 11K videos from this dataset.

- **CinePile** is a long-form video understanding dataset. The training set has 9,248 videos, from which we choose 5K videos. The videos are sourced from English-language films on the YouTube channel MovieClips, which provides self-contained clips.

# B    DETAILS OF BENCHMARKS

In this section, we will provide a detailed description of the benchmark we evaluated.

- **video-SALMONN-2 testset** comprises 483 videos spanning 14 distinct domains. Each video has a duration ranging from 30 to 60 seconds, with an average length of 51 seconds. To evaluate caption quality, a judge model is employed to process the generated caption along with the ground-truth event, which then identifies three types of errors: *Missing Events*, *Incorrect Events*, and *Hallucination Events*. The latter two are categorized as manifestations of model hallucination. The total error rate is then obtained by summing the missing rate and the hallucination rate.

- **UGC-VideoCap** consists of 1,000 short TikTok videos, each under 60 seconds in duration and containing at least one meaningful audio segment lasting no less than 5 seconds. Each video's caption is evaluated by a judge model that assigns scores on a 1-to-5 scale across three dimensions: visual, audio, and details. These dimension scores are then normalized and aggregated to produce a final caption quality score.

- **Daily-Omni** is an audio-visual question answering benchmark comprising 684 videos depicting diverse everyday life scenarios, sourced from multiple platforms. These videos are densely multimodal, offering rich visual and auditory cues. The benchmark includes 1,197 multiple-choice question-answer pairs, distributed across six core tasks. In our experimental setting, we assess the quality of generated captions by feeding them into a judge model and measuring their capacity to support accurate question answering.

- **WorldSense** exhibits a tightly integrated coupling between audio and visual modalities, demanding that models effectively harness the synergistic perceptual power of omni-modal data. The

dataset comprises 1,662 temporally synchronized audio-visual clips, systematically categorized into eight distinct semantic domains. To facilitate comprehensive evaluation, it further includes 3,172 multiple-choice question-answer pairs spanning 26 diverse downstream tasks. In our experimental framework, we evaluate the quality of generated captions by feeding them into a dedicated judge model and measuring their efficacy in enabling accurate question answering.

- **VDC** comprises 1,027 diverse videos. The captioning model is required to generate captions for each video along five distinct dimensions using five specific prompts; these five categories of captions are then fed into an evaluation model to answer questions, thereby assessing the captioning capability. In our experiments, we evaluate our model on the "detailed" subset.

- **DREAM-1K** is a challenging benchmark for detailed video description, featuring 1,000 clips from diverse sources such as films, stock footage, and short-form videos. Each video is paired with fine-grained human-annotated descriptions, and evaluated using AutoDQ, a metric better suited for assessing rich, multi-event narratives than traditional captioning scores.

## C  IMPLEMENTATION DETAILS

In the AVoCaDO SFT stage, the model is trained for 2 epochs with a batch size of 128 and a learning rate of $2 \times 10^{-5}$. During the AVoCaDO GRPO stage, training is performed for 1 epoch with a batch size of 64 and a learning rate of $1 \times 10^{-5}$. For each query, we sample 8 responses using a temperature of 1.0. The KL-divergence regularization coefficient $\beta$ is set to 0.04, which is commonly used in previous works (Feng et al., 2025a). Both the video and audio encoders remain frozen throughout training, and only the adapters and the LLM backbone are updated.

During both training and evaluation, video inputs are sampled at 2 fps, and the resolution of each frame is limited to a maximum of $512 \times 28 \times 28$ pixels. Due to the base model's context window limitation of 32K tokens, the total video tokens is restricted to $25600 \times 28 \times 28$. All training is conducted on 16 NVIDIA H200 GPUs, while evaluation is performed on NVIDIA H20 GPUs.

## D  ADDITIONAL ANALYSIS

### D.1  ANALYSIS OF THE AUDIOVISUAL VIDEO CAPTION GENERATION BY GEMINI

In Fig. 6, we compare the audiovisual captions generated directly by Gemini-2.5-Pro with those produced by the two-stage audiovisual captioning approach used in constructing our SFT dataset (Sec. 3.1). The results indicate that direct caption generation tends to omit information from either the audio or visual modality, unlike the two-stage strategy, which provides more comprehensive coverage. To ensure high data quality, we therefore adopted the two-stage captioning method for building our SFT dataset.

### D.2  ANALYSIS OF THE THRESHOLDS IN LENGTH-REGULARIZED REWARD

In this section, we detail the rationale for selecting the length thresholds $\tau_1 = 2048$ and $\tau_2 = 4096$ in the length-regularized reward $\mathcal{R}_L$ (Eq. 8). As a preliminary, it is important to note that Qwen2.5-Omni supports a maximum context window of 32K tokens and encodes audio at a rate of 25 tokens per second. In our training and evaluation, to effectively capture video dynamics and preserve the visual detail of each frame, we sample videos at 2 fps, with each frame allocated a maximum of 512 tokens for encoding. Due to the context window constraint, the total number of video tokens is capped at 25,600.

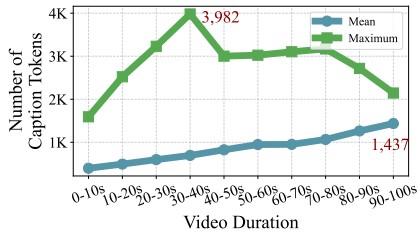

Figure 5: Distribution of caption token lengths across video durations.

The upper threshold, $\tau_2 = 4096$, is determined by the maximum feasible video duration that the model can process. Fig. 5 shows our analysis of the caption lengths generated by Gemini-2.5-Pro for videos of varying durations, which reveals that for videos up to 100 seconds, the maximum caption length rarely exceeds 3,982 tokens. A 100-second high-resolution

**Cases on Two Caption Methods**

**Direct Caption :** The video begins with a still image of a friendly-looking, purple cartoon gorilla with a pink bow on her head. She is sitting with her hands clasped, smiling and looking upwards. To her right is a white card displaying the uppercase and lowercase letter "Gg" in black. The entire image is set against a vibrant orange background with radiating lines. A female voice, presumably the gorilla's, speaks. "Hi, I'm a good gorilla. This is the /g/ sound. This is the letter G."The image then transitions. The gorilla and the letter card disappear, replaced by a blank white card with an orange border. An animation demonstrates how to write the uppercase letter "G". A voice counts "One" as a large, curved C-shape is drawn in blue. The voice then says "Two" as a short, horizontal pink line is drawn inward from the middle-right of the curve, completing the "G". Next, the animation demonstrates how to write the lowercase letter "g". The voice counts "One" as a small blue circle is drawn. The voice then says "Two" as a pink line is drawn downwards from the right side of the circle, ending in a hook to the left, completing the "g".The scene then returns to the original image of the purple gorilla next to the "Gg" card. The female voice says the sound, "/g/, /g/, good gorilla." A child's voice repeats, "/g/, /g/, good gorilla."The female voice then asks, "What's the beginning sound?" The child's voice answers, "/g/." The video ends on the still image of the gorilla and the letter card.

**Two-Stage Caption:** The scene opens with a static shot featuring a cartoon gorilla and a letter card. On the left, a friendly-looking, purple cartoon gorilla sits with its hands clasped together. It has large, expressive eyes looking upward and a gentle smile. A small pink bow is perched on its head. The gorilla and the letter card are set against a vibrant orange and yellow sunburst background, all enclosed within a thick orange border. To the right of the gorilla is a white rectangular card. The uppercase letter "G" and the lowercase letter "g" are printed in a large, black, sans-serif font on the card. A female character speaks with a high-pitched, sweet, and friendly voice. She begins by introducing herself enthusiastically, "Hi. I'm a good gorilla." Her tone then shifts to be more instructional, yet remains gentle and encouraging as she explains, "This is the 'guh' sound. This is the letter G." The scene transitions to an instructional graphic demonstrating how to write the letter "G." There are no characters in this scene. The background remains a white rectangle framed by the orange and yellow sunburst border. On the left, a completed uppercase "G" is shown in black. To its right, an animated graphic demonstrates the strokes needed to form the letter. A male narrator counts "One" as a large, curved C-shape is drawn in blue. The voice then says "Two" as a short, horizontal pink line is drawn inward from the middle-right of the curve, completing the "G". The setting is unchanged from the previous scene. The graphic for the uppercase "G" remains on the left. To the right, a completed lowercase "g" appears in black. Next to it, an animated graphic shows how to write the letter. A blue, circular shape is drawn first, followed by a curved, downward pink line to form the tail. Dotted lines and numbers again illustrate the two-step process. The narrator's voice counts again, "One. Two." The scene returns to the original shot of the gorilla and the letter card. The purple gorilla with the pink bow is shown again, sitting in the same pose with a pleasant expression. The background is the same orange and yellow sunburst pattern. The white card displaying the uppercase "G" and lowercase "g" is positioned to the right of the gorilla. The entire frame briefly fades to a lighter, washed-out color before returning to its full vibrancy. The female character's voice returns, clearly and slowly enunciating the phonetic sound, "Guh. Guh." She then says the example phrase with a cheerful lilt, "Good gorilla." A young boy's voice repeats after her, mimicking the sounds with a clear, youthful tone, "Guh. Guh. Good gorilla." Finally, the female character's voice poses a question in a friendly, engaging manner, "What's the beginning sound?" The young boy's voice confidently answers, "Guh."

Figure 6: Comparison between direct captioning and our proposed two-stage approach. Colored text highlights information present in the two-stage captions but absent in the direct captions, with audio-related and visual-related content distinguished accordingly.

video consumes 2,500 audio tokens (100s × 25 tokens/s) and the maximum 25,600 video tokens, totaling 28,100 tokens for multimodal input. When combined with the input text prompt and the generated caption, the total token count approaches the 32K context limit. To prevent context overflow and ensure the generation of complete and untruncated captions, we constrain our training dataset to videos of 100 seconds or less. Consequently, the maximum target output length, $\tau_2$, is set to 4096, providing a safe margin.

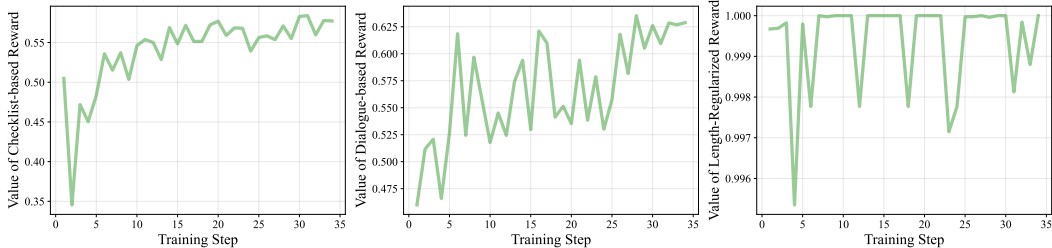

Figure 7: Reward curves during the AVoCaDO-GRPO stage.

The lower threshold, $\tau_1 = 2048$, is designed to strike a balance between comprehensiveness and conciseness for practical applications. Fig. 5 shows that the mean caption lengths for videos under 100 seconds are below 1,437 tokens. Based on this observation, we set the first threshold $\tau_1$ at 2048, a value comfortably above the average, to grant the maximum length reward to outputs of typical length. For captions with lengths between $\tau_1$ and $\tau_2$, the length reward decreases linearly. This reward structure incentivizes the model to autonomously learn a trade-off between generating a more detailed caption and optimizing other reward metrics related to factual accuracy and completeness.

### D.3 REWARD CURVES DURING TRAINING

In Fig. 7, we present the evolution of the three reward functions used during the AVoCaDO-GRPO stage. As shown, the checklist-based reward $\mathcal{R}_C$ and the dialogue-based reward $\mathcal{R}_D$ steadily increase and approach convergence throughout training. The length-regularized reward $\mathcal{R}_L$ occasionally exhibits sharp dips during training, which occur when the model encounters particularly challenging samples that induce repetition collapse in the generated caption. Notably, the minimum values of these dips gradually rise over time, indicating that the model's generation stability is improving. By jointly optimized by these three complementary reward functions, AVoCaDO is enabled to further enhance temporal coherence and dialogue accuracy while mitigating repetition collapse and effectively regulating caption length, ultimately demonstrating strong capabilities in generating high-quality audiovisual captions.

### D.4 PERFORMANCE IN MUSIC AND GENERAL SOUND SCENARIOS

We evaluate AVoCaDO in music and general sound scenarios on AVQA (Yang et al., 2022), MUSIC-AVQA (Li et al., 2022), and MUSIC-AVQA-v2.0 (Liu et al., 2024), using Gemini-2.5-Pro as the judge model to answer QA queries based on generated textual captions. The results are summarized in Tab.5. As shown, AVoCaDO not only demonstrates strong performance in video-speech related scenarios, but also exhibits significantly superior capability in describing music and general sound, substantially outperforming the baseline model Qwen2.5-Omni and approaching the performance of the commercial Gemini-2.5-Pro.

| Model | AVQA | MUSIC-AVQA | MUSIC-AVQA-v2.0 |
|---|---|---|---|
| Gemini-2.5-Pro | 72.4 | 72.8 | 50.5 |
| Qwen-2.5-Omni | 66.6 | 55.8 | 29.2 |
| AVoCaDO (Ours) | 71.8 | 62.0 | 45.8 |

Table 5: QA performance by Gemini-2.5-Pro based on textual captions in music and general sound scenarios. To mitigate answer guessing when the caption lacks necessary information, the model is instructed to refrain from answering such questions, which are then marked as incorrect.

## E FUTURE WORKS

Although AVoCaDO demonstrates substantial gains in audiovisual captioning, several promising directions remain for future enhancement: (1) detecting and mitigating hallucinations (Liu et al., 2025a; Chen et al., 2025b; Li et al., 2025a) in generated captions to improve their faithfulness and reliability; and (2) balancing the trade-off between latency and accuracy in real-time settings, for example by integrating token compression (Yao et al., 2025b; Ding et al., 2026) or related efficiency-oriented strategies to accelerate inference while preserving caption quality.

---

**Caption Comparison across Models**

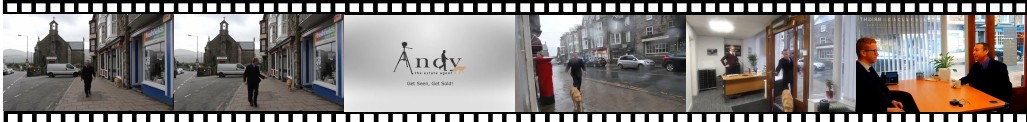

**video-SALMONN-2:** The video begins with a serene scene of a small-town street on a cloudy day… A dog leisurely follows Andy as he greets the viewers, saying, "Good morning, Andy here with Ruby and we're off to ask an expert." As the scene transitions, the logo for 'Andy the Estate Agent' appears… Andy and Ruby continue their walk, passing by the familiar storefronts, as Andy shares, "As you can see, the weather's against us but we're off inside to go and see Andy Bright at Ayles Jones and Bright Solicitors." **The narrative then moves indoors, showing Andy entering the office of Ayles Jones and Bright Solicitors through glass doors. Inside, the office is modern, with a wooden desk and a potted plant adding a touch of greenery. Another man, presumably Andy Bright, stands near the desk, engaged in a phone call. Andy greets Andy Bright warmly, "Come on, let's go in the dry," followed by, "Then we go out of here," as they proceed further into the office.** Inside, the two men sit across from each other at the desk, enjoying cups of coffee. Andy expresses gratitude towards Andy Bright, saying, "Alright, hello Andy. Thank you very much for inviting me round this morning to answer a few questions to help people." Throughout the video, the audio complements the visuals effectively, capturing the casual yet professional atmosphere of the day while highlighting the community and business interactions taking place in this charming town.

**UGC-VideoCaptioner:** The video opens with a man in a black suit and purple shirt walking down a quaint, slightly overcast street… and a small dog is seen walking alongside him. The scene then transitions to a white screen with a logo that reads "Andy the estate agent"... The next scene shows the man and his dog walking through a wet rain, and then entering a modern office with a large window showing the same street outside. Inside, he sits at a wooden desk with a young man in glasses, and they engage in a conversation, with the man in the suit speaking and the young man listening attentively. The video concludes with the man in the suit continuing to speak, presumably answering questions, while the young man listens. The overall tone is professional and informative, focusing on estate - related topics.

**AVoCaDO:** A man in a dark suit and a purple shirt walks along a brick sidewalk in a quaint town. A golden-colored dog trots beside him… The man speaks in a friendly and upbeat tone, introducing himself and his companion, "Good morning. I'm Andy Sanderson. This is Ruby, and we're off to ask an expert." A white screen appears, followed by a black-and-white animated logo. The logo features the name "Andy" in a stylized font, with the words "the estate agent" underneath. The audio transitions to a brief, modern, and slightly dramatic musical sting with a whooshing sound effect, creating a professional and contemporary feel. The scene shifts to a view from inside a building, looking out through a rain-streaked window. The same man in the suit and his golden dog are on the wet sidewalk outside. The man gestures with his arms as he walks, saying with a slightly more energetic and cheerful delivery, "As you can see, the weather's against us, but we're off inside to go and see Andre Bright at Owain and Jones and Bright Solicitors." **He then addresses his dog in a playful, encouraging tone, "Come on, Ruby, let's go in in the dry." A man in a maroon sweater stands inside at a desk, holding the telephone receiver to his ear. The first man follows up with a calm, direct instruction, "Andy, go out of here." The man and his dog walk through a glass door into the building.** Inside an office, the man in the suit sits across a wooden table from a younger man with glasses, who is wearing a dark suit jacket. **The older man** **holds a white mug and speaks with a polite tone****, "Hello, Andre. Thanks very much for inviting us round this morning to answer a few questions to help people",** **while the younger man listens attentively.** The sound of rain can be heard faintly in the background.

Figure 8: Qualitative comparison of AVoCaDO against two contemporary captioning models: video-SALMONN-2 and UGC-VideoCaptioner. Errors in baseline outputs are highlighted in red; the superior coverage and precision of AVoCaDO are highlighted in blue. **Correct** / **incorrect** **audiovisual temporal alignment** is bolded, while sound effect descriptions are underlined.

## F    ADDITIONAL QUALITATIVE RESULTS

In Figs. 8 and 9, we present qualitative comparisons of AVoCaDO against two contemporary captioning models, video-SALMONN-2 and UGC-VideoCaptioner.

**Caption Comparison across Models**

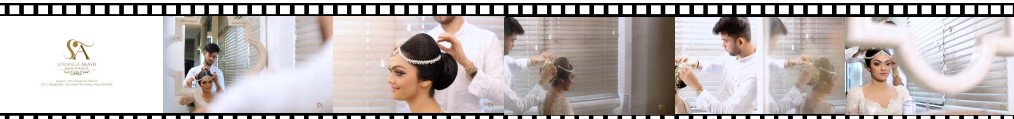

**video-SALMONN-2:** The video begins with a static image displaying the logo and contact information for 'Suranga Akash Salon Suranga.' The logo features a stylized letter 'S' with a decorative curve, accompanied by the salon's name in elegant gold text… As this introductory image fades, the scene transitions to a more intimate setting where a stylist… As the video progresses, the focus remains on the same stylist continuing his work. This time, he is adding floral decorations to the woman's updo… **Throughout these clips, the audio complements the visual narrative.** A female voice begins by saying, "A story to begin with ultimate glamour and colour. Dressing embedded with sheer joy and affection. Making the moment to remember and cherish." **The voice continues to speak softly,** enhancing the sense of anticipation and the importance of looking one's best on such a special day. The narration throughout the video maintains this tone, providing context and depth to the visual storytelling of the stylistic transformation taking place in this elegant setting. The combination of elegant visuals and subtle, meaningful audio creates a cohesive and engaging narrative that highlights the artistry and care behind every aspect of the styling process.

**UGC-VideoCaptioner:** The video opens with a white screen displaying a gold logo and text for "Suranga Akash Salon Suranga" along with contact information. The scene then transitions to a well - lit, elegant room with white blinds and a mirror, where a man in a white shirt… The video concludes with a close - up of the finished look, highlighting the elaborate headpiece… The audio throughout the video is a soft, melodic background music, which complements the visual presentation of the detailed bridal preparation.

**AVoCaDO:** A static shot displays a business card on a plain white background. The card features a gold, stylized logo of the letters "SA" with a flourish. Below the logo, the text reads "SURANGA AKASH" and "SALON SURANGA," followed by "MOBILE | 071101650, 0112504770" and an address, "75/7, BAGORU, SRI SUBUTHI ROAD, WELLAWATTE." The audio opens with a gentle, modern electronic track featuring a soft, rhythmic beat and ambient synth pads, creating a sophisticated and relaxed atmosphere. The scene shifts to a view through a decorative, star-shaped mirror. A man in a white shirt… **A male narrator with a deep, smooth, and professional voice begins to speak in a calm and measured tone, "A story to begin with ultimate glamour and color." The camera then moves to a close-up of the woman's profile… The narrator continues, "A dressing embedded with sheer love and affection, making the moment to remember and cherish." The man, now more in focus, continues to work on the woman's hair… The narrator says, "Dedicated to ensure that you look the best on your most precious day, the wedding day."** His delivery is warm and reassuring, conveying a sense of care and expertise that complements the elegant background music.

Figure 9: Qualitative comparison of AVoCaDO against two contemporary captioning models: video-SALMONN-2 and UGC-VideoCaptioner. Errors in baseline outputs are highlighted in red; the superior coverage and precision of AVoCaDO are highlighted in blue. **Correct** / **incorrect audiovisual temporal alignment** is bolded, while sound effect descriptions are underlined.

As shown in Fig. 8, video-SALMONN-2 contains multiple inaccuracies in dialogue recognition, misaligns the temporal order between the man's speech and scene transitions, and concludes with an unfitting summary. UGC-VideoCaptioner, on the other hand, omits dialogue content entirely and introduces redundant descriptions toward the end of the caption.

Similarly, in Fig. 9, video-SALMONN-2 again fails to align auditory and visual events chronologically, only mentioning the audio content at the very end of the caption. Additionally, it misidentifies the speaker's gender and overlooks the final narration segment. UGC-VideoCaptioner still neglects all spoken content, merely making a generic reference to background music at the end of the caption.

In contrast, leveraging an effective two-stage training pipeline, AVoCaDO generates high-quality audiovisual video captions that accurately synchronize audiovisual events temporally, faithfully transcribe dialogue content, and maintain strong semantic coverage in both cases.

---

**Prompts to generate video frame caption**

You are a professional video caption writer. Your task is to create a detailed, scene-by-scene narrative description of a video. For each scene, your description must include the following elements:

Main Subjects: Describe the people present, including their appearance, clothing, actions, and gestures.
Setting & Background: Detail the environment, background, and any notable objects.
On-Screen Graphics: Mention the specific content of any text, titles, or emojis that appear on the screen.
Camera Work: Note any significant camera movements like zooms, pans, or angle changes.

---

Figure 10: Prompts to generate video frame caption.

## G   DETAILS OF PROMPTS

### G.1   PROMPTS TO GENERATE CAPTIONS FOR SFT

Figs. 10 to 12 present the prompts used to generate video frame captions, audio captions, and to synthesize both, respectively, during the creation of the SFT caption data detailed in Sec. 3.1.

### G.2   PROMPTS TO DECOMPOSE CAPTIONS INTO KEYPOINTS

In Fig. 13, we present the prompt used to decompose a caption into keypoints, which is the foundation of the checklist-based reward detailed in Sec. 3.2.2.

### G.3   PROMPTS TO JUDGE KEYPOINT ACCURACY IN CAPTIONS

As illustrated in Fig. 14, we present the prompt designed to assess whether keypoints are accurately described in a caption, which is used to compute the checklist-based reward $\mathcal{R}_C$.

### G.4   PROMPTS TO EXTRACT DIALOGUES IN CAPTIONS

In Fig. 15, we present the prompt used to extract dialogues in the caption, which is the foundation of the dialogue-based reward detailed in Sec. 3.2.3.

### G.5   PROMPTS TO IDENTIFY SPEAKER SUBJECT CONSISTENCY

Fig. 16 shows the prompt to determine whether the speakers in each aligned pair refer to the same subject based on the video content, which is used to calculate the number of correctly matched speaker pairs $S_{speaker}$.

### G.6   PROMPTS TO ANSWER QUESTIONS BY TEXTUAL CAPTIONS

In Fig. 17, we provide the prompt used to assess the quality of a caption by leveraging it to answer questions, as described in Sec. 4.2.2.

## H   THE USE OF LLMS

Throughout the coding and debugging stages, we leveraged LLMs for technical guidance. Following the collaborative drafting of the manuscript, we again engaged LLMs to polish and refine its language and overall expression.

---

**Prompts to generate audio caption**

You are a professional audio caption writer. Your task is to create a detailed narrative description of an audio in the video. Your description must include the following elements:

Narration / Dialogue: Please accurately transcribe the spoken words (narration or dialogue) from the audio. In addition to the transcription, describe the speaker's tone and emotional delivery during the speech—such as whether the tone is calm, excited, hesitant, enthusiastic, serious, sarcastic, etc.—based on vocal cues like pitch, pace, volume, and emotion.
Music & Sound: Describe the background music's mood and any important sound effects.

The audio caption should be coherent and well-structured. Do not simply give the transcriptions without the speaker's tone and emotions.

---

Figure 11: Prompts to generate audio caption.

---

**Prompts to fuse the video frame caption and audio caption**

You are tasked with fusing the visual caption and audio caption into a single, coherent narrative based on the video content. Follow these strict rules:

1. Preserve every single sentence from both the visual caption and audio caption exactly as they appear.
2. Do NOT omit or delete any sentence in any way.
3. You may reorder the sentences (from both captions) to create a logical and temporally accurate sequence that reflects the video's events.
4. Ensure the integrated narrative flows naturally in time with the video, aligning visual actions with corresponding sounds or spoken content.

Verify before responding: Did I include every sentence from both captions?

Visual caption: {visual caption}
Audio caption: {audio caption}

Now generate the integrated audio-visual caption:

---

Figure 12: Prompts to fuse the video frame caption and audio caption.

---

**Prompts to decompose captions into keypoints**

You are an expert assistant designed for fine-grained audiovisual content analysis. Your task is to decompose a given video caption into a structured, comprehensive, and non-redundant inventory of distinct keypoints. Extract and categorize fine-grained keypoints from the given video caption according to the following five audiovisual-specific dimensions. Ensure the keypoints are atomic, precise, and non-overlapping.

1. Static Entity Description: Attributes and spatial configurations of relatively stationary entities. This includes people, objects, animals, and environmental elements.
2. Dynamic Action & Interaction: Motions, events, and pairwise or group interactions among entities that describe the evolving narrative.
3. Auditory Elements: All sound-related content, including speech, music, and ambient or diegetic sound effects, which is essential for holistic multimodal comprehension.
4. Spatio-temporal & Cinematography: Structural, stylistic, and temporal features of the video, including scene settings, transitions, temporal progression, and camera techniques.
5. Cross-modal Narrative Logic: High-level coherence where auditory and visual elements explicitly explain, complement, or guide each other to reveal the storyline or intent. This must involve an explicit temporal alignment between a sound and a visual event.

Output Format: You should output the keypoints in Python List Format: ["xxx", "xxx", ...]

Video Caption: {video caption}

Given the video caption, please list all the keypoints:

Figure 13: Prompts to decompose captions into keypoints.

---

**Prompts to judge keypoint accuracy in captions**

A good video caption is one that describes the various details in the video. Your task is to judge whether a video caption is good or not. You will be provided all the keypoints in the video, and also a video caption to be evaluated. You need to determine which keypoints are described correctly in the given video caption.

There are totally {# keypoints} keypoints in the video. All the keypoints will be provided in List format, i.e. ["xxx", "xxx", ...] The video caption to be evaluated will be provided as well.

Output Format:
Your output should be strict in the following Python dictionary format without anything else: {"Count of correctly mentioned keypoints": x, "Correctly mentioned keypoints": [...]}

Keypoints in the video: {keypoints}
Video caption to be evaluated: {video caption}

Given keypoints in the video and the video caption, please count the correctly mentioned keypoints and list them out.

Figure 14: Prompts to judge keypoint accuracy in captions.

---

**Prompts to extract dialogues in captions**

You are a highly skilled assistant specializing in extracting conversational dialogue from text. Your task is to carefully analyze the given description of a video and accurately identify and extract all dialogue content within it.

Please directly output the dialogue in the following format without adding any other content. If no dialogue is present, state: "None."

Dialogue format:
Speaker A Description: "Dialogue from speaker A."
Speaker B Description: "Dialogue from speaker B."
Speaker A Description: "Further dialogue..."

The description for each speaker (e.g., "Person in red dress") must align with the given description and should be simplified for brevity. The key is to be concise and clearly distinguish between speakers (e.g., "Man in red shirt" is sufficient).

Video description: {video description}

---

Figure 15: Prompts to extract dialogues in captions.

---

**Prompts to identify identify speaker subject consistency**

Given a video and several pairs of descriptive phrases about a certain subject, please help me determine whether the subjects in each pair refer to the same entity in the video.

For each pair of phrases, respond with 'Yes' or 'No', separated by a single space, without any extra characters. For example, if three pairs of phrases are provided, a valid response format would be: 'Yes No Yes'.

Descriptive phrases (each line contains a single pair): {dialogue pairs}

---

Figure 16: Prompts to identify speaker subject consistency.

---

**Prompts to answer questions based on textual captions**

You are a precise QA assistant. Your task is to answer multiple-choice questions based ONLY on the video caption provided.

Do not use any outside knowledge or assumptions—your answer must strictly reflect information from the caption. Always output only the capital letter corresponding to your choice (e.g., A, B, C, D). If the caption does not provide enough information to answer the question, output "N/A" instead.

Here is the video caption: {video caption}

Question: {question}
Choices: {choices}

---

Figure 17: Prompts to answer questions based on textual captions.

