# OpenReview forum: "AVoCaDO: An Audiovisual Video Captioner Driven by Temporal Orchestration"
_ICLR.cc/2026/Conference — ICLR 2026 Poster_

### Official Review · Reviewer_Zz2V · 2025-10-15

**Soundness:** 3
**Presentation:** 4
**Contribution:** 2
**Rating:** 6
**Confidence:** 4

**Summary:**

This paper introduces AVoCaDo, an audio-visual video captioner driven by the temporal orchestration between audio and visual modalities. AVoCaDo is built upon the Qwen2.5-Omni, which is enhanced by a two-stage post-training: 107K temporally-aligned audio-visual captions are used for SFT, followed by a reinforcement learning using three reward-based GRPO. Experiments verify the effectiveness of the proposed method.

**Strengths:**

- This paper involves a well-structured two-stage post-training pipeline including both SFT and RL.
- The proposed 107K high-quality audiovisual caption dataset used in the SFT can be a good resource for relevant topics.
- The paper is well-written, easy to follow. The proposed method is clearly described.

**Weaknesses:**

- The ablations lack evaluation on the dataset composition. Tab. 4 reports ablations only for reward terms, not for dataset subsets (TikTok-10M vs. FineVideo etc.). It is unclear whether gains stem from data diversity or model tuning.

- During the RL stage, the paper briefly mentions that 2K caption data is randomly sampled and  8 rollouts are generated for GRPO calculation. It would be better to provide ablation studies on the number of video samples and rollouts used in the RL stage.  Moreover, the visualization of the different reward types during training would also be helpful for readers.
- The literature review on audiovisual captioning could be more comprehensive, such as Fine-grained Audible Video Description (CVPR 2023), LongVALE: Vision-Audio-Language-Event Benchmark Towards Time-Aware Omni-Modal Perception of Long Videos (CVPR 2025), etc.
- I appreciate the simple, elegant, and clearly written paper. However, since there are many research papers using SFT and GRPO, I am afraid that the paper does not provide sufficiently new technical insights to the community.

**Questions:**

- What is the total training time for the SFT and RL? The calculation of the rewards also requires some time.
- How to define the five key points in the checklist for computing Reward_c? Will there be other caption cases beyond the predefined five points?

---

> ### Author Response · Authors · 2025-11-15
> **Response to the Reviewer Zz2V [Questions 1 & Kind Request for a Short Extension]**
>
> **Question 1: What is the total training time for the SFT and RL? The calculation of the rewards also requires some time.**
>
> **Response 1:** On 16 H200 GPUs, SFT takes 28 hours and RL takes 22 hours. You are correct that computing rewards requires additional time; however, based on our observations, the bottleneck in RL is not reward computation. Reward scores can be obtained rapidly through parallel API calls. In fact, **the primary bottleneck in RL lies in rollout generation and gradient updates**, particularly the former, which is also discussed in [1].
>
> [1] History Rhymes: Accelerating LLM Reinforcement Learning with RhymeRL
>
> ---
> **Kind Request for a Short Extension**
>
> Thank you very much for your thoughtful and insightful comments. We truly appreciate the time and care you have invested in reviewing our work.
>
> Due to the substantial number of additional ablation studies required to fully address your concerns, and given the current limitations in computational resources, which have led to relatively slow experimental turnaround, we are actively seeking access to additional GPU resources to accelerate this process.
>
> **We are committed to providing thorough, well-validated responses to all of your questions, and we kindly ask for your patience. We will provide comprehensive responses to all your concerns once all experiments are completed.**
>
> Thank you again for your understanding and support!

---

> ### Author Response · Authors · 2025-11-17
> **Response to the Reviewer Zz2V [Concerns 1]**
>
> **Concern 1: The ablations lack evaluation on the dataset composition. Tab. 4 reports ablations only for reward terms, not for dataset subsets (TikTok-10M vs. FineVideo etc.). It is unclear whether gains stem from data diversity or model tuning.**
>
> **Response 1:** Thank you for your insightful suggestion!
>
> To further examine whether the gains in the RL stage come from data diversity or model tuning, we have conducted two additional ablation experiments as recommended: one trained solely on a 2K subset sampled from TikTok-10M, and the other trained solely on a 2K subset sampled from FineVideo. The results are shown in the table below.
>
> | Method | video- | SALMONN-2 | testset | Daily- | Omni | |
> |-----|-----|-----|-----|-----|-----|-----|
> | | Total $\downarrow$ | Dlg. F1 $\uparrow$ | RepCol (%) $\downarrow$ | Avg. $\uparrow$ | Dlg. F1 $\uparrow$ | RepCol (%) $\downarrow$ |
> | AVoCaDO-SFT | 41.4 | 74.4 | 3.5 | 48.1 | 73.6 | 5.1 |
> | RL by Full Set-2K (**AVoCaDO**) | 37.3 | 76.9 | 0.4 | 50.1 | 76.2 | 1.0 |
> | RL by TikTok-2K | 37.8 | 76.7 | 0.4 | 49.4 | 75.9 | 1.2 |
> | RL by FineVideo-2K | 38.2 | 76.2 | 0.5 | 49.7 | 75.4 | 1.2 |
>
> We observe that using RL with a subset from a single data source leads to slightly lower performance compared to using a more diverse dataset. However, both settings still show clear improvements over the SFT-only model. This indicates that **most of the performance gains in the RL stage stem from our carefully designed reward functions tailored for audiovisual captioning, while data diversity further enhances the model’s generalization across different benchmarks to some extent**.

---

> ### Author Response · Authors · 2025-11-17
> **Response to the Reviewer Zz2V [Concerns 2]**
>
> **Concern 2: During the RL stage, the paper briefly mentions that 2K caption data is randomly sampled and 8 rollouts are generated for GRPO calculation. It would be better to provide ablation studies on the number of video samples and rollouts used in the RL stage. Moreover, the visualization of the different reward types during training would also be helpful for readers.**
>
> **Response 2:** Thank you for suggesting these more comprehensive ablation studies!
>
> In the table below, we report the results of using 1K and 3K samples for RL, as well as generating 4 and 12 rollouts per iteration under the 2K-sample setting.
>
> | Configuration | video- | SALMONN-2 | testset | Daily- | Omni | |
> |-----|-----|-----|-----|-----|-----|-----|
> | | Total $\downarrow$ | Dlg. F1 $\uparrow$ | RepCol (%) $\downarrow$ | Avg. $\uparrow$ | Dlg. F1 $\uparrow$ | RepCol (%) $\downarrow$ |
> | 2K data & 8 rollouts (default) | 37.3 | 76.9 | 0.4 | 50.1 | 76.2 | 1.0 |
> | 1K data & 8 rollouts | 38.8 | 75.3 | 0.8 | 48.4 | 74.9 | 1.4 |
> | 3K data & 8 rollouts | 37.0 | 77.1 | 0.4 | 50.3 | 76.5 | 1.0 |
> | 2K data & 4 rollouts | 40.1 | 73.6 | 1.9 | 48.8 | 74.1 | 2.6 |
> | 2K data & 12 rollouts | 37.0 | 77.2 | 0.3 | 50.4 | 76.7 | 0.9 |
>
> As shown, compared with our default configuration of using 2K samples and generating 8 rollouts per iteration, **reducing either the data size to 1K or the number of rollouts to 4 leads to clear performance degradation**. Conversely, **increasing the data size to 3K or generating 12 rollouts leads to only marginal improvements**. Considering the additional training time and computational cost required when increasing either the amount of training data or the number of rollouts, adopting 2K samples with 8 rollouts per iteration offers a more balanced choice.
>
> ---
> Thank you again for the suggestion! **We have added the training curves of the three reward types used during the RL stage to App. E.3 of the revised paper.** You may refer to the updated PDF for details.

---

> ### Author Response · Authors · 2025-11-17
> **Response to the Reviewer Zz2V [Concerns 3]**
>
> **Concern 3: The literature review on audiovisual captioning could be more comprehensive, such as Fine-grained Audible Video Description (CVPR 2023), LongVALE: Vision-Audio-Language-Event Benchmark Towards Time-Aware Omni-Modal Perception of Long Videos (CVPR 2025), etc.**
>
> **Response 3:** Thank you for the reminder!
>
> Building upon the introduction of the fine-grained audiovisual description benchmark FAVD Bench, FAVD further proposes AVLFormer, which demonstrates strong capabilities in audiovisual description. LongVALE is the first work to achieve precise fine-grained temporal video understanding in omni-modality. **We will include citations to both works in the final version of the paper.**

---

> ### Author Response · Authors · 2025-11-17
> **Response to the Reviewer Zz2V [Concerns 4]**
>
> **Concern 4: I appreciate the simple, elegant, and clearly written paper. However, since there are many research papers using SFT and GRPO, I am afraid that the paper does not provide sufficiently new technical insights to the community.**
>
> **Response 4:** Thank you for recognizing the clarity and elegance of our work!
>
> AVoCaDO focuses on the audiovisual video captioning task, and our pilot experiment highlight the importance of the temporal alignment between audio and video events in audiovisual video captioning. Building on this insight, we propose a tailored two-stage post-training strategy for generating high-quality audiovisual captions. **Our innovation does not lie in modifying the post-training paradigm itself (e.g., SFT + GRPO); rather, we introduce improvements at each stage of post-training specifically aimed at generating high-quality audiovisual captions with accurate audiovisual event alignment, significantly enhancing the model’s ability in audiovisual captioning.**
>
> Specifically, in the first stage, we collect diverse video sources and construct a carefully designed pipeline for caption generation and filtering, yielding semantically rich captions that are temporally aligned with audiovisual events. We then perform supervised fine-tuning on the model to endow it with an initial capability to generate high-quality audiovisual captions. In the second stage, leveraging reinforcement learning, we design three complementary reward functions tailored to the unique characteristics of audiovisual captions, enabling the model to iteratively self-improve toward even higher caption quality. These three rewards are: (1) a checklist-based reward to enhance semantic coverage of key audiovisual events; (2) a dialogue-based reward to improve accuracy in identifying speakers and their utterances; and (3) a length-regularized reward to normalize caption length and mitigate repetition collapse. **As indicated by Reviewer 5P4K, comprehensive ablation studies validate the effectiveness of our design.**
>
> Overall, while AVoCaDO does not modify the general post-training paradigm, it provides new task-specific insights and methodological contributions that advance the state of audiovisual video captioning.

---

> ### Author Response · Authors · 2025-11-17
> **Response to the Reviewer Zz2V [Concerns 5]**
>
> **Concern 5: What is the total training time for the SFT and RL? The calculation of the rewards also requires some time.**
>
> **Response 5:** On 16 H200 GPUs, SFT takes 28 hours and RL takes 22 hours. You are correct that computing rewards requires additional time; however, based on our observations, the bottleneck in RL is not reward computation. Reward scores can be obtained rapidly through parallel API calls. In fact, **the primary bottleneck in RL lies in rollout generation and gradient updates**, particularly the former, which is also discussed in [1].
>
> [1] History Rhymes: Accelerating LLM Reinforcement Learning with RhymeRL

---

> ### Author Response · Authors · 2025-11-17
> **Response to the Reviewer Zz2V [Concerns 6]**
>
> **Concern 6: How to define the five key points in the checklist for computing Reward_c? Will there be other caption cases beyond the predefined five points?**
>
> **Response 6:** Thank you for your insightful question!
>
> The five core dimensions in our checklist-based reward are derived **through synthesis, refinement, and careful consideration** based on two foundations: (1) existing visual-only video understanding and audio understanding frameworks, and (2) the unique requirements of cross-modal audiovisual captioning.
>
> Specifically, for **Static Entity Description**, we draw inspiration from MiraData[1], which emphasizes subject and background descriptions. We further refine “subjects” into human and object categories, and broaden “background” into environmental elements to capture richer contextual details. For **Spatio-temporal & Cinematography**, building on MiraData’s attention to camera motion, we extend this category to encompass scene transitions and temporal progression, allowing us to handle videos with more dynamic visual structures. For **Dynamic Action & Interaction**, we incorporate VidCapBench[2]’s focus on motion description, and generalize it from single-entity actions to include pairwise and group interactions among multiple entities. **Auditory Elements** aims to cover the full spectrum of possible audio content, including speech, music, and ambient or diegetic sound effects. Finally, **Cross-modal Narrative Logic** is motivated by insights from our pilot experiments, which highlighted the importance of temporal alignment between audio and video events. This dimension focuses on keypoints that the two modalities mutually explain, complement, or guide each other in conveying intent or narrative structure.
>
> [1] MiraData: A Large-Scale Video Dataset with Long Durations and Structured Captions (NeurIPS, 2024)
>
> [2] VidCapBench: A Comprehensive Benchmark of Video Captioning for Controllable Text-to-Video Generation (Findings of ACL, 2025)
>
> ---
>
> Regarding whether captions may contain information beyond these five categories, we acknowledge that due to the high diversity of real-world videos, certain extremely subtle or unusual details may fall outside the scope of the proposed checklist. However, the five dimensions we define are deliberately chosen to cover the vast majority of information essential for audiovisual understanding. **The strong captioning performance of AVoCaDO further demonstrates that these dimensions are already highly effective in guiding the model to generate semantically rich, temporally aligned, and high-quality audiovisual captions.**
>
> Importantly, our design is inherently **scalable**. Future work can further expand or refine these core dimensions within the same training framework to accommodate even more fine-grained audiovisual captioning.

---

> ### Comment · Reviewer_Zz2V · 2025-11-26
>
> Thanks for the authors' responses. My main concerns are well addressed and have no further questions to discuss. The paper is clear-written and the experimental results and analysis are comprehensive. Overall, I think this paper should be accepted. So I am happy to increase my final rating to 8. But, I encourage the authors to open-source this work.

---

> > ### Author Response · Authors · 2025-11-26
> > **Thanks for your acknowledgement!**
> >
> > Thank you very much for your positive feedback and for increasing your final rating. We truly appreciate the time and effort you spent on evaluating our work! We will open-source our work.

---

### Official Review · Reviewer_vB4b · 2025-10-31

**Soundness:** 4
**Presentation:** 4
**Contribution:** 3
**Rating:** 8
**Confidence:** 4

**Summary:**

This paper presents **AVoCaDO**, a model designed to enhance audiovisual video understanding by jointly learning from audio and visual modalities. The authors argue that most existing video captioning methods are overly vision-centric and fail to capture temporal alignment between auditory and visual events. To address this, AVoCaDO introduces a two-stage post-training framework (SFT, GRPO).
Experimental results demonstrate that AVoCaDO achieves state-of-the-art performance among open-source models on audiovisual captioning, QA, and visual-only benchmarks.

**Strengths:**

The paper is well written and easy to follow. The core idea is clearly explained, with strong logical flow from the motivation to the experimental results. Each component of the proposed approach is described with sufficient clarity, and the reasoning behind design choices is well supported by empirical evidence. The experimental results are clearly presented and convincingly demonstrate the effectiveness of the proposed method across multiple benchmarks.

**Weaknesses:**

- **Limited transparency of data construction.**
Although the paper claims to curate a 107K high-quality audiovisual dataset, the process relies heavily on closed-source models (Gemini-2.5-Pro, GPT-4.1). This limits the reproducibility and transparency of the dataset and may raise concerns about data bias or accessibility.
- **Dependence on proprietary judge models.**
The evaluation heavily depends on Gemini-2.5-Pro as a judging model, which may introduce bias or instability in QA-based evaluation. A human or open-source baseline judge would make the comparison more robust.
- **Lack of technical contribution in model design.**
While AVoCaDO is built upon Qwen2.5-Omni, the paper does not clearly describe how the audio and video inputs are encoded, temporally synchronized, and fused within the model architecture. As a result, the contribution leans more toward fine-tuning and reward engineering rather than introducing a technically novel modeling component. A clearer explanation of the underlying multimodal representation mechanism would strengthen the technical contribution of the work.

**Questions:**

Could the authors clarify whether the curated 107K audiovisual caption dataset (SFT and GRPO versions) will be released publicly or partially shared to support reproducibility?

---

> ### Author Response · Authors · 2025-11-15
> **Response to the Reviewer vB4b [Concerns 1]**
>
> **Concern 1: Limited transparency of data construction. Although the paper claims to curate a 107K high-quality audiovisual dataset, the process relies heavily on closed-source models (Gemini-2.5-Pro, GPT-4.1). This limits the reproducibility and transparency of the dataset and may raise concerns about data bias or accessibility.**
>
> **Response 1:** Thank you for your valuable suggestions!
>
> We fully understand your concern about data transparency and reproducibility. While our data construction pipeline indeed involves closed-source models, **we have provided a detailed description of the data construction process and the associated prompts in Sec. 3.1 and App. G** of our manuscript. Furthermore, to support research reproducibility and further advance the community's work in audiovisual understanding and generation, **we commit to publicly releasing the entire dataset of 107K training examples** upon paper acceptance. This ensures that future researchers can fully reproduce our results without requiring access to the proprietary models.

---

> ### Author Response · Authors · 2025-11-15
> **Response to the Reviewer vB4b [Concerns 2]**
>
> **Concern 2: Dependence on proprietary judge models. The evaluation heavily depends on Gemini-2.5-Pro as a judging model, which may introduce bias or instability in QA-based evaluation. A human or open-source baseline judge would make the comparison more robust.**
>
> **Response 2:** Thank you for your valuable suggestions!
>
> We acknowledge that model-based evaluations inevitably introduce some degree of bias. However, human evaluation also suffers from bias to some extent, and its cost is significantly higher than model-based evaluation. **Inspired by your comment, a more robust approach may be to aggregate results from multiple judge models.** In the table below, we report additional evaluation results on the Daily-Omni benchmark using two open-source judge models.
>
> | Captioner | Gemini-2.5-Pro | Qwen2.5-72B | Llama-3.3-70B |
> |-----|-----|-----|-----|
> | Qwen2.5-Omni | 13.4 | 25.3 | 14.0 |
> | UGC-VideoCaptioner | 17.0 | 27.8 | 14.2 |
> | video-SALMONN-2 | 29.9 | 35.7 | 17.2 |
> | AVoCaDO | 50.1 | 61.4 | 27.6 |
>
> As shown above, the absolute scores vary substantially across judge models. This is expected, because in QA-based caption evaluation, the judge model’s capability plays a crucial role: if the judge model is relatively weak, it may fail to answer correctly regardless of whether the caption contains a specific key piece of information. This is also one of the reasons why we use Gemini-2.5-Pro, one of the strongest widely recognized models, as our primary judge model. **Although the absolute scores differ across judge models, the relative ranking of the four captioners remains unchanged.** Therefore, aggregating evaluations from multiple judge models (or multiple human annotators) can lead to a more accurate overall ranking of captioners.
>
> ---
> **Meanwhile, we also evaluate the stability of Gemini-2.5-Pro as a judge model.** The table below reports the average score and standard deviation across three runs on the Daily-Omni benchmark, showing that Gemini-2.5-Pro exhibits strong stability as a judge model.
>
> | Captioner | Avg. | Std. |
> |-----|-----|-----|
> | Qwen2.5-Omni | 13.3 | 0.17 |
> | UGC-VideoCaptioner | 17.2 | 0.21 |
> | video-SALMONN-2 | 29.6 | 0.25 |
> | AVoCaDO | 50.0 | 0.21 |

---

> ### Author Response · Authors · 2025-11-15
> **Response to the Reviewer vB4b [Concerns 3]**
>
> **Concern 3: Lack of technical contribution in model design. While AVoCaDO is built upon Qwen2.5-Omni, the paper does not clearly describe how the audio and video inputs are encoded, temporally synchronized, and fused within the model architecture. As a result, the contribution leans more toward fine-tuning and reward engineering rather than introducing a technically novel modeling component. A clearer explanation of the underlying multimodal representation mechanism would strengthen the technical contribution of the work.**
>
> **Response 3:** Thank you for your insightful advice!
>
> **The key contribution of AVoCaDO lies in emphasizing the importance of the temporal alignment between audio and video events in audiovisual video captioning, and based on this key point, we propose a tailored two-stage post-training strategy for generating high-quality audiovisual captions.** In the first stage, we collect diverse video sources and construct a carefully designed pipeline for caption generation and filtering, yielding semantically rich captions that are temporally aligned with audiovisual events. We then perform supervised fine-tuning on the model to endow it with an initial capability to generate high-quality audiovisual captions. In the second stage, leveraging reinforcement learning, we design three complementary reward functions tailored to the unique characteristics of audiovisual captions, enabling the model to iteratively self-improve toward even higher caption quality. These three rewards are: (1) a checklist-based reward to enhance semantic coverage of key audiovisual events; (2) a dialogue-based reward to improve accuracy in identifying speakers and their utterances; and (3) a length-regularized reward to normalize caption length and mitigate repetition collapse. **As indicated by Reviewer 5P4K, comprehensive ablation studies validate the effectiveness of our design.**
>
> We select Qwen2.5-Omni as our base model due to **its audiovisual encoding and representation mechanisms are are well-suited for our requirement of generating high-quality captions with temporally aligned audiovisual events.** Below, we detail how AVoCaDO (Qwen2.5-Omni) processes audiovisual inputs.
>
> First, audio and visual inputs are encoded separately by their respective encoders and then augmented with Time-aligned Multimodal RoPE, assigning identical absolute temporal timestamps to audio and video tokens that correspond to the same timestamps. These audio and visual tokens are subsequently grouped into chunks of 2-second intervals, with video tokens placed before audio tokens within each chunk. This arrangement ensures that, at a macro level, audio and visual tokens are interleaved in the input sequence, enabling the model to jointly perceive multimodal information during sequence modeling. **This structural design directly satisfies the intrinsic requirement for generating high-quality captions with temporally aligned audiovisual events without necessitating further architectural modifications to the original model**. Only by leveraging a relatively lightweight post-training approach, AVoCaDO successfully achieves high-quality, temporally coherent audiovisual captioning.

---

> ### Author Response · Authors · 2025-11-15
> **Response to the Reviewer vB4b [Concerns 4]**
>
> **Concern 4: Could the authors clarify whether the curated 107K audiovisual caption dataset (SFT and GRPO versions) will be released publicly or partially shared to support reproducibility?**
>
> **Response 4:** We are delighted to announce that the entire 107K training dataset will be publicly released after paper acceptance, to support research reproducibility and further advance the community's work in audiovisual understanding and generation!

---

> > ### Comment · Reviewer_vB4b · 2025-11-24
> >
> > Thank you for the thorough rebuttal. The authors have addressed all of my concerns clearly, and the clarifications resolved the issues I raised in the initial review. I am satisfied with the revisions and will keep my score at 8.

---

> > > ### Author Response · Authors · 2025-11-24
> > > **Thanks for your acknowledgement!**
> > >
> > > Thank you very much for acknowledging our rebuttal! We sincerely appreciate the time and thoughtful consideration you devoted to reviewing our work and offering such constructive feedback. We’re pleased that our clarifications helped address all your concerns adequately, and we deeply value your positive assessment.

---

### Official Review · Reviewer_5P4K · 2025-11-02

**Soundness:** 3
**Presentation:** 3
**Contribution:** 3
**Rating:** 8
**Confidence:** 5

**Summary:**

The paper proposes an audiovisual captioner built on Qwen2.5-Omni with a two-stage post-training pipeline: SFT on 107K curated, temporally aligned AV captions created via a two-stage prompting scheme, and RLHF using checklist rewards combined with a dialogue reward and a length regularizer to curb repetition and excessive verbosity. The paper shows strong results on video-SALMONN-2 testset, UGC-VideoCap, and caption-driven QA transforms of Daily-Omni and WorldSense, plus competitive performance on VDC-Detailed (visual-only). Ablations studies show the effectiveness of proposed rewards and the value SFT data to the model.

**Strengths:**

1. The main novelty comes from the usage of checklist rewards and dialogue-based reward proposed in RLHF stage, and it proves very effective according to the ablation results (shown in Table 7).

2. The value of the constructed dataset is proven to be very effective across both video captioning and joint audio video QA benchmarks.

3. The paper is very well-written with clear motivation and clear analysis of the results both quantitatively and qualitatively.

**Weaknesses:**

1. The paper only showcases its strong capability in video-speech related scenarios, but audio could be from also other categories, such as music / general sound. It would be great to incorporate some experiments covering these aspects as well, which will make the contribution even stronger. Some related works in this area for references such as [1]-[3].

2. How does the method perform when the sound / speech is very noisy scenario, does the visual help answering these questions? How is the captioning like in audio-only mode in these benchmarks? It would be more comprehensive if the authors incorprate these results to see the value of vision part only.

3. How does the proposed method generalize to long context audio-visual video captioning and how does different sampling strategies in audio / video affect the final performance.

4. Can such framework work in real-time mode and what are current limitations / trade-offs between latency and accuracy? Maybe some dicussions or future works could be mentioned to showcase the challenges / limitations of existing framework.

[1] Learning to Answer Questions in Dynamic Audio-Visual Scenarios, CVPR 2023.

[2] Tackling data bias in MUSIC-AVQA: Crafting a balanced dataset for unbiased question-answering, WACV 2024.

[3] AVQA: A Dataset for Audio-Visual Question Answering on Videos, ACM MM 2022.

**Questions:**

See weaknesses.

---

> ### Author Response · Authors · 2025-11-15
> **Response to the Reviewer 5P4K [Concerns 1]**
>
> **Concern 1: The paper only showcases its strong capability in video-speech related scenarios, but audio could be from also other categories, such as music / general sound. It would be great to incorporate some experiments covering these aspects as well, which will make the contribution even stronger.**
>
> **Response 1:** Thank you for your valuable suggestions!
>
> You are absolutely correct that music / general sound constitutes an essential part of audiovisual content. **In the construction of our SFT data and the design of the checklist-based reward, we explicitly incorporated scenarios involving music and general sound**, enabling the model to learn to recognize and describe them accurately. For instance, in **Figure 4**, AVoCaDO identifies *“a sharp, percussive sound effect of hands slapping the table”* in the first line, and later produces a detailed description of *“an upbeat, energetic, and slightly retro-sounding musical jingle with a female vocalist singing 'Food Mania Review'”* in the fifth line. **These two examples directly demonstrate AVoCaDO’s capability to accurately characterize general sound and music.**
>
> ---
>
> Moreover, **we have also evaluated our model on the three listed benchmarks using caption-driven QA**, with the following results:
>
> | Model | AVQA | MUSIC-AVQA | MUSIC-AVQA-v2.0 |
> |-----|-----|-----|-----|
> | Gemini-2.5-Pro | 72.4 | 72.8 | 50.5 |
> | Qwen2.5-Omni | 66.6 | 55.8 | 29.2 |
> | AVoCaDO | 71.8 | 62.0 | 45.8 |
>
> AVoCaDO not only demonstrates strong performance in video-speech related scenarios, but also **exhibits significantly superior capability in describing music and general sound**, substantially outperforming the baseline model Qwen2.5-Omni and approaching the performance of the commercial closed-source model Gemini-2.5-Pro. **We will cite these benchmarks and provide further further discussion in the final version of the paper.**

---

> ### Author Response · Authors · 2025-11-15
> **Response to the Reviewer 5P4K [Concerns 2]**
>
> **Concern 2: How does the method perform when the sound / speech is very noisy scenario, does the visual help answering these questions? How is the captioning like in audio-only mode in these benchmarks? It would be more comprehensive if the authors incorprate these results to see the value of vision part only.**
>
> **Response 2:** Thank you for your insightful question!
>
> Unfortunately, we haven't found any benchmark specifically designed to evaluate sound or speech recognition accuracy under noisy scenarios. We believe this is an interesting research direction for future work.
>
> Nevertheless, to investigate the performance of AVoCaDO in noisy conditions, we manually select 50 videos with noticeable background noise from the video-SALMONN-2 testset. Their video IDs are:
>
> ["10", "11", "110", "112", "117", "119", "12", "126", "129", "13", "133", "136", "137", "138", "140", "143", "148", "149", "15", "150", "153", "154", "155", "157", "166", "171", "179", "182", "192", "195", "197", "202", "204", "209", "213", "214", "218", "22", "220", "222", "227", "234", "236", "237", "238", "24", "240", "242", "250", "258"]
>
> We then evaluate the model’s captioning performance on this noisy subset. The results are shown below:
>
> | Dataset | Miss $\downarrow$ | Hall. $\downarrow$ | Total $\downarrow$ |
> |-----|-----|-----|-----|
> | full set | 21.1 | 16.2 | 37.3 |
> | noisy subset | 23.0 | 15.2 | 38.2 |
>
> As the table shows, **the performance on the noisy subset does not degrade significantly, demonstrating the robustness of AVoCaDO in noisy environments**.
>
> ---
> To further explore the importance of the visual modality, we have also tested our model under even more extreme conditions, namely, the visual-only setting, to evaluate the value of the vision part. Additionally, we also test AVoCaDO’s performance under audio-only mode, as shown in the table of results on the video-SALMONN-2 testset below:
>
> | Input Modality | Miss $\downarrow$ | Hall. $\downarrow$ | Total $\downarrow$ |
> |-----|-----|-----|-----|
> | audio & visual | 21.1 | 16.2 | 37.3 |
> | visual only | 34.1 | 17.4 | 51.5 |
> | audio only | 46.6 | 19.3 | 65.9 |
>
> The results demonstrate that when only the visual modality is provided, the model’s audiovisual captioning capability degrades significantly; however, the degradation is even more severe under audio-only input. This phenomenon can be attributed to the fact that **both audio and visual modalities are indispensable components for audiovisual captioning. Providing only one modality inevitably compromises the completeness of the generated captions**, as reflected by the higher Miss rate.
>
> The reason audio-only input leads to greater performance degradation is that, for a given duration of audiovisual content, **the visual modality is generally considered to contain richer information than the auditory modality**, which is further supported by the larger number of visual tokens compared to audio tokens. Consequently, **across the vast majority of scenarios, the visual modality plays a more critical role than the audio modality**.

---

> ### Author Response · Authors · 2025-11-15
> **Response to the Reviewer 5P4K [Concerns 3]**
>
> **Concern 3: How does the proposed method generalize to long context audio-visual video captioning and how does different sampling strategies in audio / video affect the final performance.**
>
> **Response 3:**
> Thanks for your insightful question again!
>
> Regarding long context audiovisual video captioning, as detailed in App. E.2, due to the maximum context length of 32K supported by our base model, Qwen2.5-Omni, we constrained the training and evaluation videos to a maximum duration of 100 seconds. However, we are pleased to report that AVoCaDO exhibits strong generalization capabilities beyond this limit. For instance, on the WorldSense benchmark, where 749 out of 1,662 videos (45%) exceed 100 seconds, **AVoCaDO achieves a substantial improvement of 41.2% over the previous best open-source model (18.2 → 25.7), despite not being explicitly trained on such long videos.** However, the absolute performance on long-form videos still has considerable room for improvement. In future work, we plan to explore several promising directions, including extending the model’s context window, employing token compression techniques for audiovisual representations, or leveraging hierarchical temporal modeling to better handle extended durations while preserving comprehensive and accurate captions.
>
> ---
>
> Regarding how does different sampling strategies in audio / video affect the final performance, we have done experiments on the video-SALMONN-2 testset and the results are shown below.
>
> | Sampling Strategy (A & V) | Miss $\downarrow$ | Hall. $\downarrow$ | Total $\downarrow$ |
> |-----|-----|-----|-----|
> | 16 kHz & 2 FPS (default)| 21.1 | 16.2 | 37.3 |
> | 16 kHz & 1 FPS | 25.1 | 15.4 | 40.4 |
> | 16 kHz & 3 FPS | 23.0 | 15.9 | 38.9 |
> | 8 kHz & 2 FPS | 26.5 | 17.4 | 43.8 |
> | 24 kHz & 2 FPS | 23.3 | 16.7 | 40.1 |
>
> The results indicate that reducing either the audio sampling rate or video frame rate leads to a notable performance drop, as critical auditory or visual information is lost, which are essential for high-quality captioning. However, increasing the sampling rate slightly degrades performance as well. This is likely because both pre-training and our post-training stages used fixed configurations of 16 kHz audio and 2 FPS video, making the model less adept at handling denser input. Moreover, it's worth noting that, under a fixed context-length constraint, sampling more video frames inevitably reduces per-frame resolution, a trade-off between frame rate and spatial fidelity that we aim to investigate further to enable comprehensive and accurate captioning of long videos.

---

> ### Author Response · Authors · 2025-11-15
> **Response to the Reviewer 5P4K [Concerns 4]**
>
> **Concern 4: Can such framework work in real-time mode and what are current limitations / trade-offs between latency and accuracy? Maybe some dicussions or future works could be mentioned to showcase the challenges / limitations of existing framework.**
>
> **Response 4:** Yes, **the proposed framework can operate in real-time mode**. Our implementation is built upon Qwen2.5-Omni, which is explicitly designed for streaming scenarios and supports low-latency incremental inference. In real-time settings where latency is critical, **the main trade-off arises from the input image resolution for each frame**. Since visual tokens produced from video frames dominate the computation for long-context multimodal processing, using lower-resolution frames is often necessary to reduce the computational overhead and improve response time. However, this inevitably leads to a drop in fine-grained captioning accuracy, as low-resolution inputs may blur subtle visual details that are important for precise recognition and temporal grounding.
>
> We will add a dedicated discussion in the Challenges section to highlight this latency–accuracy trade-off. In addition, our Future Work section will incorporate directions such as efficient visual / audio token compression (e.g., TimeChat-Online [1]), which could further reduce latency without significantly compromising captioning quality.
>
> [1] TimeChat-Online: 80% Visual Tokens are Naturally Redundant in Streaming Videos, ACM MM 2025.

---

> ### Author Response · Authors · 2025-11-15
> **Response to the Reviewer 5P4K [Overall]**
>
> Finally, **we sincerely thank you for all your valuable experimental suggestions!** We will carefully incorporate each of your proposed experiments and their corresponding insights into the final version of our paper, aiming to provide meaningful empirical experience for the audiovisual captioning community.

---

> > ### Comment · Reviewer_5P4K · 2025-11-26
> >
> > Thanks authors for providing detailed analysis and results regarding my questions and concerns! Please incorporate these additional materials in the final version of the paper. In general, I think the paper is very solid so I keep my rating as accept.

---

> > > ### Author Response · Authors · 2025-11-27
> > > **Thanks for your acknowledgement!**
> > >
> > > Thank you very much for recognizing the solidity of our work! We sincerely appreciate the time and effort you devoted to reviewing our manuscript, as well as your recommendation to accept it. We will carefully incorporate the additional analyses and results you suggested into the final version. Thank you once again for your thoughtful feedback and valuable time.

---

### Official Review · Reviewer_yLZR · 2025-11-04

**Soundness:** 3
**Presentation:** 3
**Contribution:** 3
**Rating:** 6
**Confidence:** 3

**Summary:**

This paper proposes an audiovisual video captioning system that generates temporally-aligned descriptions. The approach employs a two-stage training pipeline: (1) AVoCaDO SFT fine-tunes Qwen2.5-Omni on 107K synthetic audiovisual captions, and (2) AVoCaDO GRPO applies reinforcement learning with three complementary rewards targeting temporal coherence, dialogue accuracy, and caption quality. The paper is well-motivated by a compelling pilot study and demonstrates strong empirical results across multiple benchmarks.

**Strengths:**

1. The pilot experiment (Figure 1) provides compelling evidence that joint audiovisual captioning with temporal alignment significantly improves understanding.
2. The topic itself is of high value. Temporally-aligned audiovisual captions are important for training next-generation video understanding and generation models.
3. Well-designed reward functions for AV captioning.
4. The proposed method shows strong improvement over the base model

**Weaknesses:**

Comparing transcription accuracy against specialized ASR models (e.g., Whisper) would be useful to understand the quality of the model caption. Metrics like Word Error Rate (WER) on the dialogue portions could quantify transcription quality.

**Questions:**

N/A

---

> ### Author Response · Authors · 2025-11-15
> **Response to the Reviewer yLZR**
>
> **Concern: Comparing transcription accuracy against specialized ASR models (e.g., Whisper) would be useful to understand the quality of the model caption. Metrics like Word Error Rate (WER) on the dialogue portions could quantify transcription quality.**
>
> **Response:** Thank you for your valuable suggestion!
>
> Since the video-SALMONN-2 testset provides rich and accurate ASR ground-truth annotations, we conduct our experiments on this benchmark. For the transcripts directly generated by Whisper, no additional processing is required. In contrast, for the ground truth and the captions produced by AVoCaDO, which contain detailed audiovisual information, we use Gemini-2.5-Pro to extract the speech-only content, and manually spot-check the results to verify the reliability of this extraction procedure. The table below presents a comparison of transcription quality between AVoCaDO and four Whisper models of different sizes.
>
> | Model | WER $\downarrow$ |
> |-----|-----|
> | AVoCaDO | 17.9 |
> | Whisper-large-v3-turbo | 16.6 |
> | Whisper-medium | 17.0 |
> | Whisper-small | 18.0 |
> | Whisper-base | 21.0 |
>
> The results indicate that **although AVoCaDO is designed as a comprehensive audiovisual captioning model rather than a specialized ASR system, it still achieves competitive transcription accuracy.** Its ASR accuracy is only slightly behind the most powerful specialized ASR models, Whisper-large-v3-turbo and Whisper-medium, while outperforming Whisper-small and Whisper-base.

---

### Author Response · Authors · 2025-11-19
**Gentle Reminder on Rebuttal Review**

Dear Reviewers,

We hope you are doing well. We would like to kindly follow up and ask whether you could take a moment to review our rebuttal and share any updated assessments when convenient. Your time and effort are greatly appreciated, and please let us know if any further clarification would be helpful.

Best regards,

Authors of Submission 700

---

### Meta-Review · Area_Chair_a3Un · 2026-01-07

**Summary:**

Reviewers found the paper to be well motivated and clearly written with strong performance across multiple audiovisual captioning benchmarks and a well-designed reward-based post-training strategy. The main concerns raised in the initial reviews focused on 1) the scope of experimental validation beyond the video-speech scenario (e.g., music and general sounds), 2) robustness and modality contribution analyses (e.g., noisy conditions, audio-only vs. visual-only), 3) ablation studies on dataset composition and reinforcement learning configurations, and 4) questions regarding reproducibility and technical novelty given reliance on closed-source models.

**Reviewer Concerns:**

***Concerns Adequately Addressed by the Rebuttal***

Evaluation scope and robustness (Reviewers 5P4K, yLZR):
The authors added extensive new experiments covering music and general sound scenarios, noisy audio conditions, audio-only and visual-only settings, long video contexts, and sampling-rate trade-offs.

Ablation studies and training configuration (Reviewer Zz2V):
The rebuttal includes thorough ablations on dataset composition, RL sample size, rollout numbers, and reward behavior during training, directly addressing concerns about the source of performance gains.

Reproducibility and reliance on closed-source models (Reviewer vB4b):
The authors clearly committed to publicly releasing the full 107K dataset and provided additional evaluations using multiple open-source judge models.

Technical clarity and architectural questions (Reviewer vB4b):
The authors clarified how audiovisual inputs are temporally aligned and processed within the base model, making the technical design and modeling assumptions sufficiently clear.

ASR comparison (Reviewer yLZR):
The authors provided a direct comparison with Whisper models using WER, showing competitive transcription quality despite not being an ASR-focused model.

***Outstanding Concerns***

Limited technical contribution (Reviewers vB4b, Zz2V):
While the rebuttal clearly positions the contribution as task-specific insights and post-training design rather than architectural innovation, some reviewers may still view the technical novelty as incremental. However, this is largely a matter of framing rather than a remaining technical gap, and does not detract from the demonstrated effectiveness of the approach.

Overall, no major technical concerns remain unresolved.

**Reviewer Scores:**

Reviewer yLZR (initial score: 6):

Likely to increase to 7 or 8, as their sole concern regarding ASR comparison was directly addressed.

Reviewer 5P4K (initial score: 8):

Explicitly confirmed acceptance and would maintain score at 8.

Reviewer vB4b (initial score: 8):

Explicitly stated that all concerns were resolved and would maintain score at 8.

Reviewer Zz2V (initial score: 6 → updated to 8):

Already increased score to 8 after reviewing the rebuttal and additional experiments.

***Overall, the post-rebuttal consensus among reviewers strongly favors acceptance.***

---

### Decision · Program_Chairs · 2026-01-26

Accept (Poster)